# Demonstration of intracellular real-time molecular quantification via FRET-enhanced optical microcavity

Yaping Wang[1,7], Marion C. Lang[1,6,7], Jinsong Lu[1,7], Mingqian Suo[1], Mengcong Du[1], Yubin Hou[2,3,4,5], Xiu-Hong Wang ® [1,3,4,5] ✉ & Pu Wang[2,3,4,5]

Single cell analysis is crucial for elucidating cellular diversity and heterogeneity as well as for medical diagnostics operating at the ultimate detection limit. Although superbly sensitive biosensors have been developed using the strongly enhanced evanescent fields provided by optical microcavities, real-time quantification of intracellular molecules remains challenging due to the extreme low quantity and limitations of the current techniques. Here, we introduce an active-mode optical microcavity sensing stage with enhanced sensitivity that operates via Förster resonant energy transferring (FRET) mechanism. The mutual effects of optical microcavity and FRET greatly enhances the sensing performance by four orders of magnitude compared to pure Whispering gallery mode (WGM) microcavity sensing system. We demonstrate distinct sensing mechanism of FRET-WGM from pure WGM. Predicted lasing wavelengths of both donor and acceptor by theoretical calculations are in perfect agreement with the experimental data. The proposed sensor enables quantitative molecular analysis at single cell resolution, and real-time monitoring of intracellular molecules over extended periods while maintaining the cell viability. By achieving high sensitivity at single cell level, our approach provides a path toward FRET-enhanced real-time quantitative analysis of intracellular molecules.

Molecular studies at single cell resolution have received growing attention in the last decade due to the increasing awareness of intrinsic cellular heterogeneity of gene and protein expression as well as metabolites production[1–3]. The big challenge of single cell analysis, however, is the extremely low numbers of molecules, i.e., an average of only $1 \times 10^5$ molecules (nM level) for proteins. Current single cell techniques, such as laser scanning confocal imaging, surface-enhanced Raman spectroscopy, as well as the sequencing techniques for "omics",

albeit play crucial roles in obtaining substantial molecular information, there are limitations in respect to real-time and quantitative analysis. Thus, the combination of efficient sample manipulation and highly sensitive detection is urgently desired.

Optical microcavities, such as microspheres[4–6], microrings[7–9], microdisks[10,11], and similar configurations[12,13], which confine light within a small cavity, generating whispering gallery modes (WGMs) due to total internal reflection, have demonstrated great capability in

[1]Laboratory for Biomedical Photonics, Institute of Laser Engineering, Faculty of Materials and Manufacturing, Beijing University of Technology, 100124 Beijing, China. [2]Laboratory for Advanced Laser Technology and Applications, Faculty of Materials and Manufacturing, Beijing University of Technology, 100124 Beijing, China. [3]Key Laboratory of Trans-scale Laser Manufacturing Technology, Ministry of Education, Beijing, China. [4]Beijing Engineering Research Center of Laser Technology, Beijing, China. [5]Beijing Colleges and Universities Engineering Research Center of Advanced Laser Manufacturing, Beijing, China. [6]Present address: Carl Zeiss Microscopy GmbH ZEISS Group, Kistlerhofstr.75, 81379 Munich, Germany. [7]These authors contributed equally: Yaping Wang, Marion C Lang, Jinsong Lu. ✉e-mail: wxh2012@bjut.edu.cn

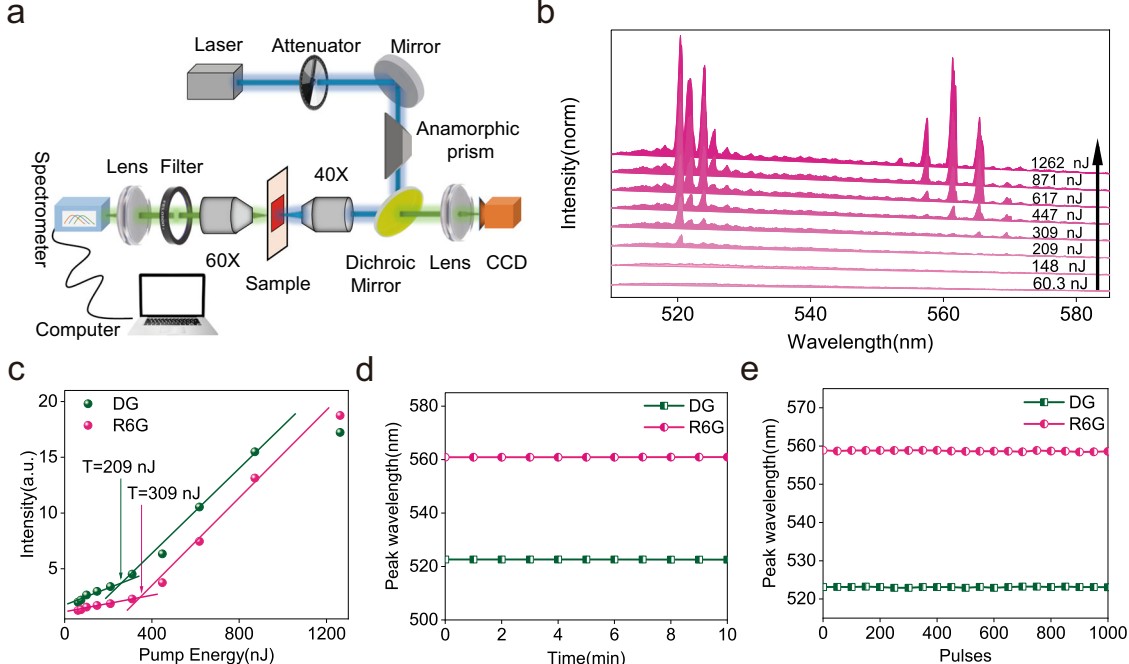

**Fig. 1 | Dual-lasing of DG and R6G via an optical microcavity. a** Schematic representation of the optical set-up (for details please see Methods section). **b** Typical lasing spectra of a DG microsphere in 50 μM R6G under varied pumping energy from 60.3nJ to 1262nJ. **c** Lasing thresholds of DG (green) and R6G (pink) are at 209nJ and 309nJ, respectively. **d**, **e** Lasing frequencies of DG (green) and R6G (pink) are independent of pumping duration (**d**) and number of pulses (**e**).

bioanalysis due to their high quality factor, ultra-low loss, ultra-long photon lifetime and ultra-high intracavity power and intensity[14–16]. Compared to most single-pass optical devices, such as waveguides[17] and optical interferometers[18], where light interacts with the analyte only once; in WGM based approaches, light can interact with the analyte as often as $10^5$ times[16], greatly enhancing the sensing performance. In order to achieve ultrahigh sensitivity or ultrafast sensing, several mechanisms have been explored, including evanescent coupling of the microcavity resonance to a plasmonic nanoantenna[19–21], laser-frequency locking[22], exceptional point technique[23] and others. Microcavity-based optical sensors can resolve single molecules or particles showing promise in the detection and manipulation of viruses, proteins and antibodies for clinical diagnostics and environmental monitoring[15,24–28]. Recently, the reach of evanescent and plasmonic techniques have extended the detection limit down to single biomolecules with dimensions in the single nanometer range[15,25]. However, these studies with optical WGM microcavity biosensors are technically not applicable for intracellular applications due to the adoption of fiber or prism couplers[29–31].

A promising solution is the active mode WGM microcavity with gain material, such as a dye-doped or intrinsically luminescent optical micro-resonator. While the small volume enables embedding into a cell, it also allows free space excitation with the pump light without adoption of a coupler. It has been shown that WGMs can be successfully created from small dye-doped fluorescent beads (with sizes ranging from sub-micrometer to micrometer, depending on the refractive index of the bead material) that are taken up by individual cells, to oil droplets stained with dye injected into a cell[32–34]. Lasers embedded in the cytoplasm of a cell or tissue, have recently been employed to record transient cardiac contraction profiles with cellular resolution as well as for high-density optical barcoding of cells[35–37]. The intracellular microcavity interacts with molecules in the cytosol via evanescent mode coupling and can serve as an optical sensor to detect molecules of interest. However, intracellular sensing of a molecule-of-interest has not been feasible so far due to the extreme low quantity of cytosolic molecules at

single cell level. Here, we demonstrate that incorporating Förster resonance energy transfer (FRET) to an active-mode WGM microcavity platform can greatly increase the sensitivity enabling intracellular quantitative sensing of small molecules at single cell resolution.

## Results and discussion

### FRET-assisted dual lasing of a micro-resonator

First, lasing action from a dye-doped microsphere was evaluated before introducing FRET to the lasing line. A dragon green (DG, MW 369 Da) doped polystyrene (PS) microspheres (1% DG doping, $n = 1.59$) with diameter of 15 μm ($D$) was ingrained in ultrapure water (ddH$_2$O) and excited by a 473 nm pulsed diode-pumped laser (pulse duration 2.5 ns, repetition rate 100 Hz, diameter of pump spot on sample 30 μm). A typical WGM emission pattern was observed (Fig. S1a) with a center wavelength (CWL) at 534 nm. The center peak consistently remains the maximum height in the spectral envelope under optimized excitation conditions, so it can be easily identified without ambiguousness. The lasing threshold was 26.3 nJ. Above this threshold, the emission intensity increases linearly with the pumping energy (Fig. S1b). The free spectral range (FSR) of DG obtained from the spectrum was 3.79 nm, which is in good agreement with the theoretical estimation of 3.81 nm (Fig. S1c).

Subsequently, a DG microbead was embedded in 50 μM rhodamine 6 G (R6G) solution and pumped with the same laser. The optical setup is shown in Fig. 1a. After overcoming the lasing threshold, two subsets of lasing peaks were observed: one group of peaks with CWL around 520 nm, attributed to DG emission; and a second group of peaks with CWL around 565 nm, attributed to R6G emission (Fig. 1b). The lasing outputs of the two sets of peaks depend on the pumping power. As shown in Fig. 1c, the relationship between output intensity and pump energy exhibits a characteristic s-shaped curve, similar to Fig. S1b. The lasing thresholds are 209 nJ and 309 nJ for DG and R6G, respectively. During 10 min of continuous laser operation, or $1 \times 10^3$ pump pulses, the lasing frequency remained unshifted (Fig. 1d, e), determined by peak fitting to the lasing spectra ($n = 3$), although the

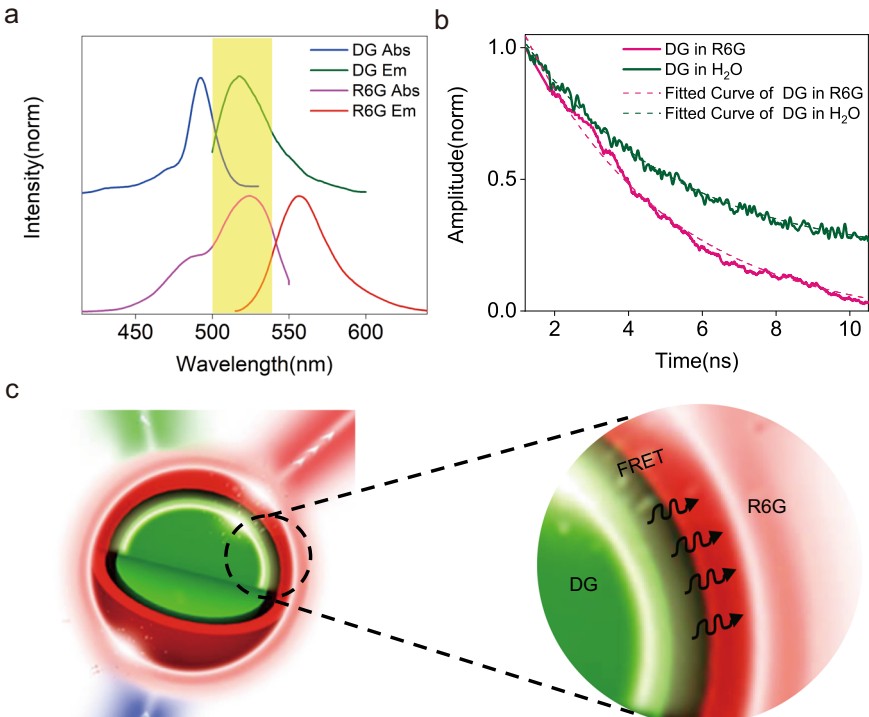

**Fig. 2 | FRET determines R6G lasing. a** Absorption and emission spectra of DG and R6G show a strong overlap of DG emission and R6G absorption spectra, indicating good energy conservation. **b** Time-resolved fluorescence decay of DG in the absence (green) and presence (pink) of R6G. The shorter fluorescence lifetime of DG in the presence of R6G indicates FRET from DG to R6G. **c** Schematic illustration of energy transfer via FRET at the interface of the microcavity.

output intensities of DG and R6G decreased over time due to photo bleaching (Fig. S2a, b).

As a control experiment, a sodalime glass microsphere ($n = 1.59$, $D = 15\,\mu m$)) without fluorophore-doping was embedded in $50\,\mu M$ R6G and excited under the same conditions. In this case, we only observed broad fluorescence emission of R6G, with a maximum @558 nm (Fig. S2c). Increasing the R6G concentration to $500\,\mu M$ and pump energy to as high as 10,000 nJ resulted in intensified fluorescence (Fig. S2c, d), however, did not generate lasing output. This control experiment indicates that R6G lasing depends on the DG emission.

The lasing action of R6G can be a consequence of two possible mechanisms. First, the DG lasing output functions as a pumping source to excite the R6G molecules; or secondly, Förster resonance energy transfer (FRET) between DG and R6G takes place[38]. To test the first hypothesis, a $15\,\mu m$ sodalime glass microsphere was embedded in the same R6G solution, and a 532 nm nanosecond-laser was used as a pumping source, which provides a similar wavelength to DG lasing but with much higher single-pulse energy. However, only spontaneous emission rather than lasing of R6G was observed (Fig. S3a), ruling out the first hypothesis (Here we would like to clarify that with the described setup, R6G lasing can be realized, however, much higher R6G concentration (>5 mM) and pumping energy (>mJ) are required, please see Fig. S3b, c). The second option, FRET, requires a strong overlap between the emission spectrum of the donor, DG, and the absorption spectrum of the acceptor, R6G, to ensure energy conservation, which is clearly fulfilled for DG and R6G (Fig. 2a). A second prerequisite for FRET is the distance between donor and acceptor, meaning that if the distance between DG and R6G is within 1–10 nm, efficient energy transfer via FRET can occur. As the polystyrene microsphere is negatively charged, the cationic dye R6G would be readily adsorbed on the anionic surface, thus meeting the distance requirement for FRET (c.f. SI Fig. S4 for charging properties of the microspheres and molecules). Therefore, presumably, the two fluorophores form an

ideal FRET pair, in which DG acts as donor (D) and R6G acts as acceptor (A).

To investigate this hypothesis further, we analyzed the fluorescence decay curves of DG in the presence of R6G, as the FRET process would affect the fluorescence decay of the confined donor DG in the presence of a suitable acceptor. A time-correlated single-photon counting (TCSPC) was applied to evaluate the fluorescence decay curve after excitation with a short pulse of light. The fluorescence decay of DG in the absence of R6G ($\tau D$) was determined to be 4.12 ns (mean value) regardless of the location of ROI (region of interest), Fig. S5 (DG-H$_2$O). In contrast, in the presence of R6G, the decay time ($\tau DA$) of DG located on the surface of the microsphere was reduced to 2.14 ns (Fig. 2b, and Fig. S5 DG-R6G), whereas the decay time for the interior DG of the microsphere remains unchanged (4.10 ns, Fig. S5). Considering the high spectral overlap between the dyes, clearly, the energy transfer mechanism from surface DG to R6G was further evidenced as Förster type via dipole-dipole coupling. The energy transfer process is illustrated in Fig. 2c. The un-altered $\tau$ value of interior DG molecules (Fig. S5) indicates no energy transfer occurred in this situation.

FRET efficiency ($E_{FRET}$) was determined to be 48% ($E_{FRET} = (\tau D-\tau DA)/\tau D$)[39], which is high compared with the published data[40–42]. The ultra-high FRET efficiency is very likely owing to the evanescent field of WGM, as evidenced in the literature that an evanescent field would enhance the fluorescent energy transfer[43,44]. Due to the adjustment in spatial distance between donor and acceptor molecules and the variation of R6G molecular orientation, FRET efficiency may vary. Indeed, we obtained different $\tau DA$ values (Fig. S5) for different ROIs on the microsphere surface, corresponding to $E_{FRET}$ of 52%, 47% and 42%, 43% and 59%, respectively. The data indicate that R6G lasing is a joint result of FRET and WGM.

By measuring the FSRs of DG and R6G on the observed spectra at different R6G concentrations, we found FSR$_{DG}$ is always smaller than FSR$_{R6G}$ (c.f. Table S1). We also calculated the cavity sizes for DG and R6G using $FSR = \lambda 2/n\pi D$. The resonant cavity for R6G is constantly

larger than that of DG (c.f. Table S2). The decreased FSR of DG can be contributed to the increased refractive index near the sphere surface. Due to the layer of R6G, the DG observes a little bit refractive index increase. Whereas for R6G, since, somehow, the gain position is a little outside, its WGM might be pulled outward slightly.

As described previously[16,45], for a reactive WGM sensor, by first order perturbation theory, the frequency shift ($\Delta\omega$) can be estimated by Eq. (1):

$$\frac{\triangle\omega}{\omega} = -\frac{\alpha_{ex}|\mathbf{E}(r_0)|^2}{2\int\varepsilon|\mathbf{E}(r)|^2dV} \tag{1}$$

where $\varepsilon$ is the permittivity of the medium and $\alpha_{ex}$ is the polarizability of the particles (molecules) bound to the microcavity; $\mathbf{E}(r_0)$ and $\mathbf{E}(r)$ are the modal field amplitudes at the binding site $r_0$ and throughout the mode, respectively. The frequency shift produced by a molecule binding to the microcavity is proportional to the intensity $\sim\mathbf{E}^2(r_0)$ encountered at the binding site $r_0$. Therefore, once a molecule is bound on the surface of a WGM microcavity where the evanescent field strength $\mathbf{E}(r)$ is high, the molecule will become polarized at the optical frequency $\omega$. The energy that is needed to polarize the molecule and induce the dipole moment is $\frac{1}{2}\alpha_{ex}\mathbf{E}(r_0)^2$. Since FRET enables effective energy transfer from DG molecules to R6G molecules, located on the binding site $r_0$, which contributes extra energy $\mathbf{E}_x(r_0)$ on top of the evanescent field. Any mechanism that can amplify the field intensity at the binding site will dramatically increase the sensitivity in molecule detection, as evidenced by the plasmonic effect in the field[46,47]. Therefore, we hypothesize that FRET-coupled WGM (FRET-WGM) would exhibit higher sensing performance than its non-FRET counterpart (non-FRET-WGM).

## Resonant energy transfer greatly enhances WGM sensing performance

To test this hypothesis, we compared the sensing performance of FRET-WGM and non-FRET-WGM. First, a DG microsphere was embedded in various concentrations of R6G solution and pumped with a 473 nm laser. Depending on the concentration of R6G, different resonance shifts were observed (Fig. 3a, only the center peaks are shown here. for the full spectra please refer to SI figure S6a, b). As the concentration of R6G increased, the CWL of DG lasing shifted toward shorter wavelengths (blue shift), i.e., from 534 nm in pure water to 520 nm in 50 μM R6G solution. Correspondingly, the CWL of R6G lasing shifted toward longer wavelengths (red shift) (Fig. 3a).

By measuring eleven different concentrations (0, 10 nM, 50 nM, 100 nM, 500 nM, 1 μM, 5 μM, 10 μM, 50 μM, 100 μM and 500 μM) and repeating the measurement for five times, the mean value of resonance wavelengths, $\lambda_{DG}$ and $\lambda_{R6G}$, were taken to draw the concentration-dependent curve (Fig. 3b). The average wavelength distances between CWLs of DG and R6G ($\Delta\lambda = \lambda_{R6G}-\lambda_{DG}$) at different R6G concentrations show an exponential increase with increase of R6G concentration. The log-scale curves are shown in Fig. 3c (for a mathematical model please see Table S3). The inset shows linear fit to the curve. For a R6G concentration of 500 μM, the wavelength distance $\Delta\lambda$ is 53.5 nm; a considerable distance of 12.8 nm is already observable at a R6G concentration of 1 μM; and for 10 nM R6G, the $\Delta\lambda$ value is 6.2 nm. Thus, the observed changes in frequency shift could be a sensitive means to quantify the extra-cavity acceptor molecules in solution. The calculated limit of detection (LOD) for R6G is 15.2 pM (we define the LOD as equal to the linewidth FWHM ($\delta\lambda$), i.e., $\Delta\lambda_{LOD}=\delta\lambda$), opening up the potential to detect very small concentration changes of the acceptor dye with high sensitivity.

In the second experiment, we addressed the FRET effect on WGM sensing. Therefore, a mirror-study of "non-FRET-WGM" was performed. To minimize errors caused by non-FRET factors, such as molecular size and dipole moment, a non-fluorescent R6G analog, R6G

hydrazide (R6GH), was synthesized[48], utilizing the unique property of rhodamine transformation from the fluorescent ring-opened form (R6G) to the non-fluorescent spirolactam (R6GH) (Fig. 4a). While R6GH and R6G both are positively charged (Fig. S4) and have similar molecular size (MW 457 vs 479 Da, confirmed by ESI-MS, Fig. 4a right panel, for the full mass spectrum refer to Fig. S7) and polarity, R6GH molecules do not absorb or emit energy in the visible region (Fig. 4b). Therefore, there is no energy transfer between DG and R6GH. Time-resolved fluorescence decay also confirmed this ($\tau$ values of DG in the presence of R6GH remains same as in H$_2$O regardless of the location of the measurement, Fig. S5 DG-R6GH). A DG microsphere was ingrained in various dosages of R6GH solution and excited with a 473 nm laser. In clear contrast to the aforementioned FRET-WGM results, the presence of R6GH molecules resulted in a dose-dependent red-shift of the DG resonance wavelength (Fig. 4c, for full spectra please refer to SI fig. S6c). From curve fitting of the $\Delta\lambda$-concentration plot (Fig. 4d), the LOD for R6GH was calculated to be 747.4 nM (again using $\Delta\lambda_{LOD}=\delta\lambda$, see table S3 for detailed mathematical model). The comparison of DG-R6G with DG-R6GH shows that the evaluated $LOD_{FRET-WGM}$ is around $5\times10^4$ times lower than that of the non-FRET-WGM system. Clearly, introducing FRET to the active-mode WGM sensing system greatly increases the sensing performance.

Comparing the obtained spectra at the same concentration of R6G and R6GH additionally shows that the lasing threshold of $DG_{FRET-WGM}$ is much higher than that of $DG_{non-FRET-WGM}$ (Fig. 4e). This finding further indicates that energy in fact is transferred from DG to R6G, increasing the losses for the DG WGM system. An increased concentration of R6G also leads to a higher lasing threshold of DG (Fig. 4f), since more energy is transferred from DG to R6G, causing more losses for the DG system.

It is interesting to note the switch of the resonance wavelength shift of DG ($\lambda_{DG}$) when the concentrations of R6GH and R6G increase, respectively, from a red-shift in the R6GH-case to a blue-shift in the R6G-case, indicating different underlying mechanisms. In the non-FRET-WGM system, as documented in many publications, the shift to longer resonance wavelength occurs because the bound R6GH molecules will effectively "pull" part of the optical field to the outside of the microsphere by $\Delta l$ [49], thereby increasing the roundtrip path length by $2\pi\Delta l$. This increase in path length produces the shift ($\Delta\lambda$) to lower frequencies (red-shift). The shift of the resonance wavelength mainly depends on the refractive index change. For the investigated R6G concentrations, the change of the refractive index ($\Delta n$) is small (c.f. SI table S4), corresponding to a small shift of the resonance wavelength $\Delta\lambda$. This explains the much lower sensitivity of a pure active WGM.

In the case of FRET-WGM, the refractive index change of R6G ($\Delta n$) is equally small to that of non-FRET-WGM (c.f. SI table S4). However, the exceptionally high FRET efficiency suggests FRET plays the dominating role in the sensing process, overcompensating the effect by the refractive index change, although the mechanism is not yet fully understood at the moment. For more comprehensive insights into energy transfer of the microcavity, we applied rate equations to simulate the intensity for every single mode and thereby obtain the efficiency of cavity energy transfer, please refer to SI section 3. Analyzing each mode of the rate equations reveals that FRET effect is sufficient to achieve population inversion, that is to say, FRET dominates the process for R6G lasing. With FRET being the dominating factor, the mechanisms of the active-mode WGM system investigated here are fundamentally different from the mechanisms underlying a non-FRET-WGM system. The described FRET-WGM configuration allows energy transfer from inner cavity to out cavity, distinguishing from intra-cavity FRET reported previously[50–53].

To validate the theoretical model, we altered the surface charge of the microsphere to abolish the distance requirement for FRET to occur. To this end, we used PAH molecule (Poly (allylamine hydrochloride), MW 10,000–20,000 Da) to coat the microsphere to achieve a positively charged surface. TEM image confirmed a ~5 nm layer

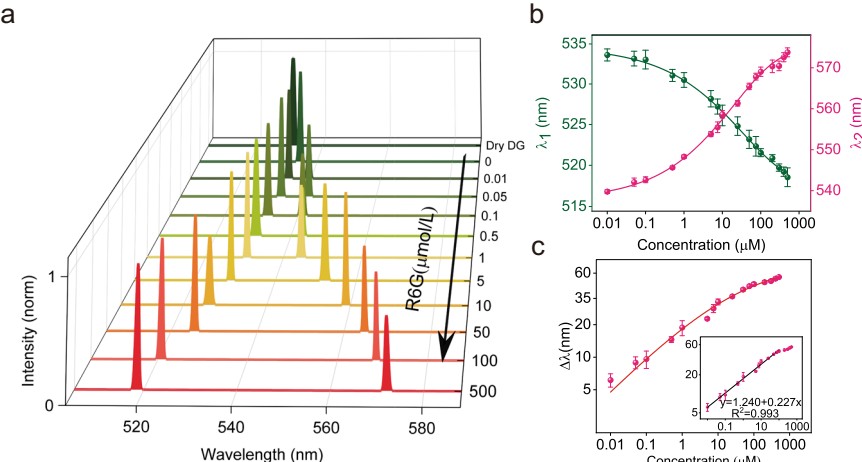

**Fig. 3 | The FRET-WGM sensing system. a** Dose-dependent lasing spectra (only showing the CWL) of DG and R6G displaying wavelength blue-shift of DG and red-shift of R6G with increase of R6G concentration. **b** Mean wavelength-concentration plot of DG (green) and R6G (pink) of five experiments. Data are presented as mean values ± SD ($n = 5$). **c** log-scale Δλ-concentration curve showing that the wavelength gap (Δλ) exponentially increases as the R6G concentration increases (for a mathematical model ref. SI table S3, $\Delta\lambda = \lambda_{2(R6G)} - \lambda_{1(DG)}$). Data are presented as mean values ± SD ($n = 5$).

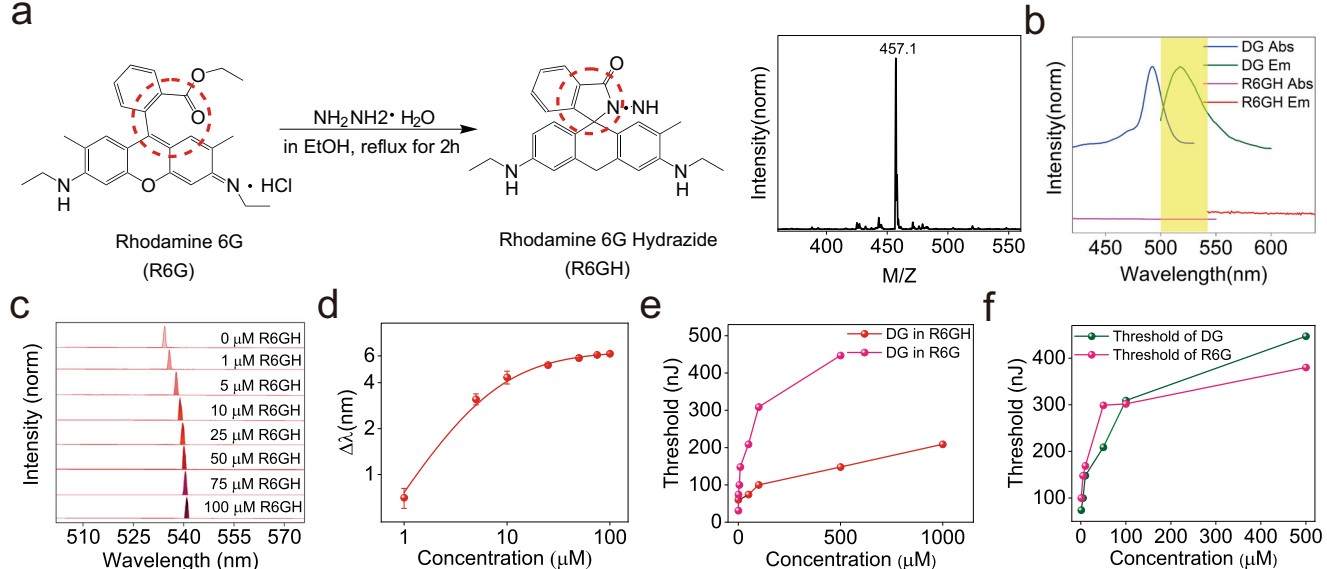

**Fig. 4 | The non-FRET-WGM sensing system. a** Structural formula of R6G and R6GH. R6GH is a R6G analog with similar structure and molecular size. Right panel shows the molecular weight of R6GH determined by ESI mass spectrometer (cf. Fig S8 for full mass spectrum). **b** Absorption and emission spectra of DG and R6GH. Unlike R6G, R6GH does not absorb and emit light in the visible region. **c** Dose-dependent lasing spectra of DG in the presence of various concentrations of R6GH (solvent is 90%$H_2O$ + 10%EtOH. Only center peaks are shown, full spectra in

SI Fig. S6c). **d** Wavelength shift (Δλ)-concentration curve of DG in the presence of R6GH. $\Delta\lambda = \lambda_{DG-in-R6GH} - \lambda_{DG-in-solvent}$. Data are presented as mean values ± SD ($n = 3$). **e** Threshold of DG in the presence of R6G (pink) or R6GH (red). Due to FRET, the lasing threshold of DG in R6G is much higher than that of R6GH. **f** Lasing thresholds of DG and R6G. As the R6G concentration increases, the thresholds of DG rise, indicating an increased energy loss of DG molecules.

formed around the microsphere (Fig. S8a, b). The measured zeta potential of the DG microspheres after PAH modification was 15.9 mV (Fig. S8c). The positively charged microsphere was embedded in 50 µM R6G solution and pumped with 473 nm pulsed laser. We randomly selected 5 microspheres and the data are shown in Fig. S8d. We observed solely DG lasing instead of DG and R6G dual lasing. The CWL of DG lasing spectrum is 535.8 nm, which is comparable with that in pure H2O (red-shifted by 1.8 nm due to the increased refractive index caused by the PAH modification). It is assumed that the positively charged DG microsphere repels R6G molecules and thus R6G molecules are distanced from the microsphere, interfering with the energy transfer via FRET, as a result, R6G fails to lase. The data further support

the mechanism of FRET playing a leading role in the FRET-WGM sensing system.

In the FRET-WGM format, the shift of DG toward shorter wavelengths indicates that with increased R6G concentration, the energy transfer from DG to R6G becomes more effective, leading to higher losses in the DG lasing process and a higher gain in the R6G lasing process. The FRET-WGM is analogous to a quasi-3-level lasing system, where increased losses lead to a frequency shift toward shorter wavelengths[54]. This is because increased losses require a higher gain and subsequently a higher excitation level. This in turn means the maximum gain of the system is at a shorter wavelength. This is what can be observed for the DG lasing: higher R6G concentration increases

**Table 1 | Δλ (λ_{R6G}-λ_{DG}) values at same R6G concentration with different DG-doping**

| R6G Conc. | 10 nM | 50 nM | 100 nM | 500 nM | 1 μM | 5 μM | 10 μM | 25 μM | 50 μM | 75 μM | 100 μM | 200 μM | 300 μM | 400 μM | 500 μM |
|---|---|---|---|---|---|---|---|---|---|---|---|---|---|---|---|
| Δλ(nm) (1%DG) | 6.15 | 8.93 | 9.63 | 14.59 | 18.81 | 20.81 | 32.53 | 36.54 | 42.29 | 45.52 | 47.50 | 49.49 | 50.75 | 53.36 | 55.23 |
| Δλ(nm) (2%DG) | 3.56 | 8.68 | 11.93 | 17.17 | 20.85 | 26.93 | 35.61 | 41.44 | 45.23 | 47.71 | 49.92 | 53.35 | 55.41 | 56.75 | 57.07 |

the loss for DG lasing, because more of the energy is transferred to the R6G transition, leading to a shift toward shorter excitation wavelengths. The R6G emission, on the other hand, shifts to longer wavelengths, because here the gain is increased at higher concentrations. Supporting this explanation is that with increasing R6G concentration also a higher lasing threshold for DG lasing was observed (Fig. 4f). The relationship between wavelength gap (Δλ) and acceptor concentration can be further quantitatively simulated via absorption/emission cross section analysis (please refer to SI section 4).

The initial concentration of DG will determine the absolute value of CWL and sensitivity to R6G concentration. Increasing DG concentration in the microsphere to 2.0% ($w/w$), similar lasing performance was observed. The CWL of 2% DG microsphere in $H_2O$ is 539 nm, red-shifted by 5 nm due to higher gain molecule quantity in the resonator. Embedding the microsphere in various concentrations of R6G and pumping with a 473 nm laser, similarly, we observed dose-dependent blue- and red-shift of DG and R6G CWLs, respectively (Fig. S9a, b). At the same R6G concentration, the wavelength distance between DG and R6G ($Δλ = λ_{R6G}-λ_{DG}$, Fig. S9c) is larger for 2% DG-R6G than that of 1% DG-R6G, as shown in Table 1. The $LOD_{FRET-WGM}$ of the 2% DG microcavity sensor, which is calculated to be 3.2 pM, is lower than that of 1% counterpart (15.2 pM). The data, on the one hand, further confirmed the aforementioned results; more importantly, also provide a valid path to optimize the sensitivity of the FRET-WGM platform. In order to rule out detrimental effects by the ambient temperature, the frequency difference between DG and R6G lasing at higher temperature (30 °C) was investigated. Increasing temperature resulted in an overall blue shift of both DG and R6G resonant peaks, however Δλ ($λ_{R6G}-λ_{DG}$) value remained unchanged (Fig. S10), compared to the room temperature (20 °C) data.

The magnitude of the resonant wavelength shift Δλ ($Δλ/λ = Δω/ω$) is inversely proportional to the mode volume $V_{mode}$ given by the denominator in Eq. (2)[55]:

$$\frac{\triangle\lambda_r}{\lambda_r} = \frac{\alpha_{ex}\sigma}{\varepsilon_0(n_s^2 - n_m^2)R} \tag{2}$$

where $n_s$ and $n_m$ are the refractive indices of the sphere and exterior medium, respectively, and $\sigma$ is the surface density of bound biomolecules. Therefore, reducing the size (modal volume) of the optical resonator would further boost the sensing capability. Meanwhile, usage of a high $Q$ microcavity would also increase the sensitivity of a FRET-WGM microlaser.

## FRET-WGM microlasers for intracellular molecule sensing

For intracellular operation of the FRET-WGM microlasers, a DG bead was first internalized into a cell (T47D, human breast cancer cell line) via simple co-culture and endocytosis. The 3D confocal image of the cell confirms the DG microbead was truly inside the cell rather than sitting on top of the cell (Fig. S11a and supplementary video). Most beads enter cells following 4 h of incubation. Cells harboring beads grow and divide as usual (Fig. S11b, c). A single cell harboring one DG bead was excited with 473 nm pulsed laser. We observed typical WGM emission of DG with CWL @ 537 nm (Fig. S12a), which is red-shifted by 3 nm compared to that in $H_2O$ due to the higher refractive index of cytoplasm ($n_{cytoplasm}$ 1.37 vs $n_{H_2O}$ 1.33). The

intracellular lasing threshold (60.3 nJ, Fig. S12b) is also higher than in $ddH_2O$ (Fig. S1b).

For the measurement of intracellular R6G concentration, 100 μM R6G solution was co-cultured with T47D cells harboring DG microspheres for 30 mins to allow R6G molecules to enter the cells. R6G molecules passively penetrated the plasma membrane and dispersed in the cytoplasm (Fig. 5a). A single cell harboring a microsphere was pumped with 473 nm laser. In the presence of intracellular R6G, two subsets of WGM lasing peaks could be observed (Fig. 5b, c), which is similar to the data obtained in the extracellular setup. During 10 mins of continuous laser operation or $1 \times 10^3$ pump pulses, the lasing frequency remained un-shifted (Fig. 5d, e), however, the lasing intensities of DG and R6G decreased over time due to photo bleaching (Fig S12c, d). The lasing thresholds of DG and R6G are higher than the non-cellular setup (Fig. 5f vs Fig. 1c) due to more energy loss in cell.

This finding demonstrates an advantage of using the wavelength rather than lasing intensity as a measure to quantify the concentration (although in fact, data acquisition is completed within seconds, far before severe photo bleaching occurs). Fluorescence intensity often is a measure for cross-examining cellular events and pathophysiologic conditions in small animal models of human diseases. While intensity measurements are convenient in the laboratory, they are often inadequate and sometimes imprecise in real-world situations. It is impossible to know the probe concentration at each point of the image. In addition, fluorescence intensity changes may be due to photo bleaching, photo-transformation and/or diffusive processes and so forth. On the contrary, molecular quantification using lasing wavelength can provide unique or complementary information, which is more accurate since it is not influenced by photo bleaching.

By measuring the wavelength gap between DG and R6G (Δλ) (Fig. 5g), the intracellular concentration of R6G after 5 mins incubation can be determined via the standard concentration-Δλ curve shown in Fig. 3c (Although the intracellular refractive index is slightly higher than pure water, Δλ would not be affected. Therefore, the standard curve is also applicable for the intracellular study). The measured Δλ value is 40.74 nm, corresponding to an intracellular R6G concentration of 46.12 μM (Fig. 5g, h). The discrepancy between the intracellular and the extracellular R6G concentration is due to the barrier function of the cell membrane. Similarly, 10 μM and 50 μM of R6G solution were incubated with T47D cells harboring DG microspheres, the intracellular concentrations of R6G after 5 mins incubation were determined to be 1.54 μM and 18.55 μM, respectively (Fig. 5g, h). The cells remained alive after pumping experiments were completed (cells on the slide grow and proliferate normally).

In summary, we presented a FRET-enhanced active-mode WGM sensing platform, which allows free space excitation and enables sensitive intracellular detection of R6G at single cell resolution, providing a method for fast (no amplification is needed) and cost-effective real-time molecular analysis at single cell resolution. FRET has been used extensively throughout biology, materials science, chemistry, suggesting a great potential for studying molecular interactions, energy transfer and conservation. WGM lasers offer multiple options via altering the gain medium configuration, including doped in the resonator and cross-linked on the surface of the resonator.

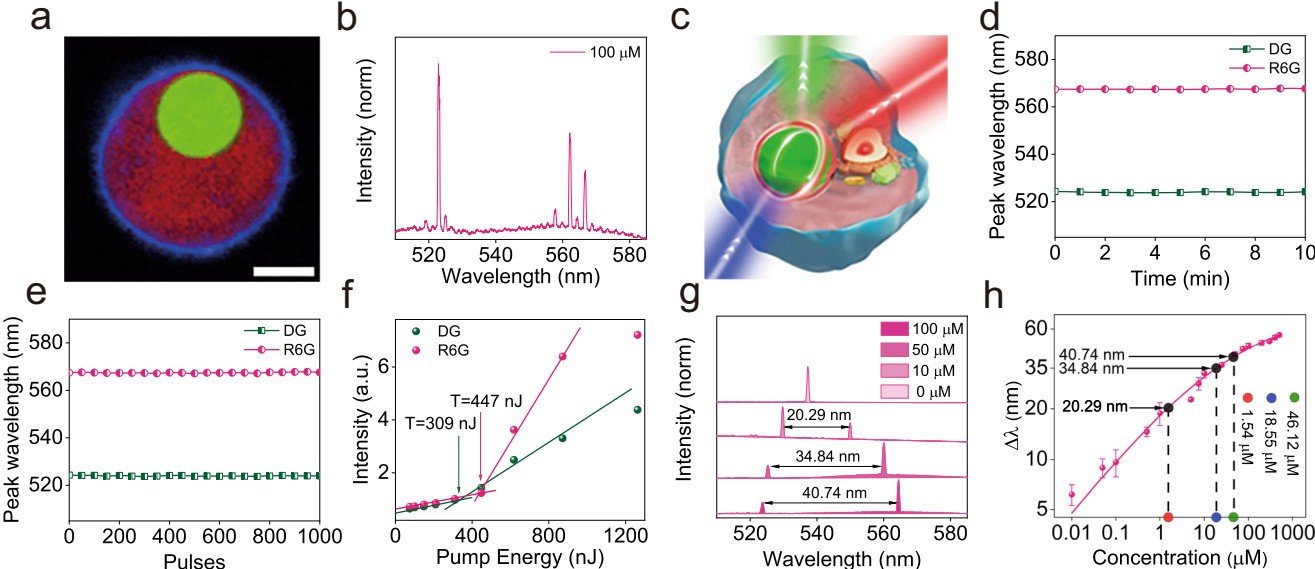

**Fig. 5 | Real-time intracellular R6G quantification. a** Image of A T47D cell harboring a DG (microsphere (green) following incubation with R6G (red) for 5 mins. The cell membrane (blue) was stained with CellMask™. The experiment was repeated independently three times with similar results. Scale bar 10 μm. **b** Dual lasing of DG and R6G within a T47D cell. **c** Schematic illustration of intracellular dual lasing via FRET from a cell harboring a FRET-donor doped optical microcavity and implanted in cytoplasm containing a FRET-acceptor. **d, e** Lasing wavelengths of DG and R6G remain un-shifted following 10 mins of continuous pumping (**d**) and 1000 pulses (**e**). **f** The intracellular lasing thresholds of DG and R6G are 309 nJ and 447 nJ, respectively, higher than that of the extracellular setup (Fig. 1c). **g** Real-time measurement of Δλ. Intracellular lasing spectra of DG and R6G are recorded following 5 mins incubation with 10 μM, 50 μM and 100 μM R6G solutions, respectively. The experiments were conducted 3 times with similar results, as-shown is a representative. **h** Intracellular R6G quantification using Δλ-concentration standard curve. The determined intracellular R6G concentrations following 5 mins incubation of R6G are 1.54, 18.55 and 46.12 μM, respectively. Data are presented as mean values ± SD (*n* = 3).

The presented approach combining FRET and WGM could be used for studying real-time molecular interactions or sensitive detection, either extracellularly or intracellularly, by means of a suitable acceptor-donor combination and WGM format. Fig. S13 gives an example for potential intracellular sensing applications. Instead of the DG-R6G combination, more cell-friendly FRET pairs, e.g., CFP-YFP, which can be co-expressed with the host protein in cytoplasm, could be considered in the future. To increase the sensitivity and reduce the mode volume, microcavities fabricated with higher refractive index materials, such as aqueous-stable perovskites, luminescent semi-conductor materials, OLED and so forth, could be employed.

Intracellular sensing to reveal the real-time information at single cell resolution, such as protein/protein (or protein/DNA, DNA/DNA) interactions, regulations of signaling molecules upon stimuli and aberrant expression under pathological conditions, can bring fundamental information and understanding of biological processes in health and disease. It also enables novel diagnostics and precise interventions for treating diseases like cancers and diabetes. The FRET assisted WGM platform provides an operative approach for realization of non-destructive intracellular sensing at single cell resolution. Further optimizing the detection limit and smart designs of FRET-WGM sensing probe to provide real-time intracellular dynamic information will be embarked in future.

## Methods
### Optical set-up
The setup is based on a 473 nm microchip laser (BrightSolutions Co. Model FP2-473-10-0.1, 473 nm 10μJ pulse energy, pulse duration 2.5 ns, repetition rate 100 Hz) for exciting WGM. The beam shape is elliptical, so a plastic prism (Thorlabs Inc.) was used to expand the smaller axis and get a better illumination of the back aperture of the objective. The laser was then focused on the sample via an objective lens (Thorlabs Inc., 40×, NA = 0.6) (Fig. 1a). The pump laser was focused to a 30 μm

large spot and a maximum pulse energy of 1–50 nJ was used depending on resonator size and tissue scattering. Emission from the microlaser was collected by the same objective, separated from the pump light by a dichroic mirror (Thorlabs Inc. Transmission band: 505–800 nm, Reflection band: 380–475 nm) and passed to the camera port of the microscope. Subsequently the fluorescence was collected via a second objective (Thorlabs Inc. 60×, NA = 0.85), the excitation light is blocked by a long pass filter (Thorlabs Inc., 490 nm) and the signal was detected by a spectrometer (Zolix Instruments Co., Omni-λ500i, resolution = 0.4 nm, 100 ms acquisition time). In the backward direction, the microbead is imaged onto a webcam to control the position of the microbead and make sure that there is only one microbead in focus for the measurements.

Laser threshold characteristics were acquired on the same set-up by varying the pump power with a set of neutral density filters. Spectra were integrated over 800 pump pulses below the threshold, whereas between 100 and 200 pump pulses were used above the threshold. 100 spectra were analyzed for each pump energy.

### Cell culture and assays
Breast cancer cell lines MCF7 and T47D were purchased from the American Type Culture Collection (ATCC). All cells were cultured in DMEM medium (Thermo Scientific) supplemented with 10% FBS, 1% Glutamax, 1% penicillin and 1% streptomycin in an incubator with 5% $CO_2$ and 80% humidity at 37 °C. Cells in the logarithmic growth phase were used for all experiments. To count the number of cells, a Countesst automated cellcounter (Invitrogen, USA) was used.

For viability assay, cells were incubated in a 96-well plate (5000 cells per well, in triplicate) with appropriate culture medium for 24 h. Subsequently, the initial medium was replaced with fresh medium containing the various concentrations of microbeads and incubated for another 24 h. The culture medium was gently removed and the cells were washed twice with sterile PBS. Then 10 μl cell counting kit-8 (CCK-8) solution was added to each well, the absorption (OD) at

450 nm was measured with the microplate reader after a 3 h co-cultivation. The following formula was used to evaluate cell viability: Cell viability (%) = (mean of Abs. value of treatment group/mean Abs. value of control) × 100%.

## Sample preparation for optical experiments

**Preparation of Poly-d-lysine coverslips.** Glass coverslips were wipe-cleaned with ethanol and left until dry. 20 µl of Poly-d-lysine (PDL, Sigma-Aldrich) was added on the top of the coverslip; the liquid was allowed to spread out to cover the entire coverslip. The samples were left at room temperature for 10 mins to ensure PDL molecules were fully adhered on the coverslip surface. Then the excessive PDL was removed and the coverslips were washed 3× with ddH2O. The coverslips were stored at room temperature on a dry place over night.

**Preparation of DG microbeads samples.** On the top of the PDL coverslip, 20 µl dragon green (DG) microbeads suspension (1:20 dilution in ddH2O) was added, after 10 mins, which allowed the microbeads to settle down, the sample was covered with a clean coverslip, and excess liquid was carefully removed. The sample was mounted on a glass slide and sealed with nail polish. After the nail polish was dry, the glass slide was mounted on a holder and the sample was fixed in the measurement setup.

**Preparation of cells harboring DG microbeads.** A clean 18 × 18 mm coverslip was placed in a 35 mm cell culture petri dish. A mixture of 2 µL DG microbeads ($10^3$ beads µl$^{-1}$) and 300 µL log-phase growing cells ($10^2$ cell µl$^{-1}$) was gently dropped on the coverslip to ensure microbeads and cells evenly cover the slip. The coverslip was stored in the $CO_2$ incubator for 4 h to allow the cells adhere on the coverslip and the microbeads to be internalized into the cells. Then the coverslip was washed three times with phosphate buffered saline (PBS) to remove un-internalized microbeads.

For intracellular real-time R6G quantification, two coverslips with live cells harboring DG microbeads were prepared as aforementioned. On the top of the coverslip, 15 µL 50 µM R6G solution was added and the samples were incubated at 37 °C for 5 mins. One coverslip was washed with PBS and the cell membrane was stained with DP for confocal imaging (please see SI for details). The other coverslip was washed and covered with a clean piece of coverslip, then mounted on a glass slide as described above and sealed with nail polish.

Additional experimental methods please refer to SI.

## Reporting summary

Further information on research design is available in the Nature Portfolio Reporting Summary linked to this article.

# Data availability

The authors declare that the main data supporting the findings of this study are available within the paper and its Supplementary Information files. Raw data used in this study are available from the corresponding author upon request.

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

## Acknowledgements

This research was supported in part by the National Natural Science Foundation of China (No: 92053116 X.H.W., 62035002 P.W., 61905006 Y.H.), the National Research and Development Program of China (No. 2017YFB0405200 P.W.), and Beijing Natural Science Foundations (L182011 X.H.W., 4192013, X.H.W.). The authors thank Dr. Yan Yinzhou at BJUT and Dr. Yao Haizi at Fuzhou University for valuable comments on the manuscript and BJUT Core Facilities for technical support. Part of this work was performed at the Technology Centre for Protein Sciences (TCPS) in Tsinghua University and Health Science Centre in Peking University.

## Author contributions

X.H.W. and M.C.L. proposed the idea and designed the experiments. M.C.L., Y.W., J.L. conducted the measurements. M.C.L, Y.W., J.L., and X.H.W. wrote the paper. X.H.W. performed modifications of the paper. X.H.W. and J.L. answered the questions raised by the reviewers. M.S., M.D., and Y.H. participated in the data analysis and discussions. X.H.W. and P.W. supervised the project.

## Competing interests

The authors declare no competing interests.
