## [Peer Review File · Nature Communications]

Demonstration of intracellular real-time molecular quantification via FRET-enhanced optical microcavityREVIEWER COMMENTS

Reviewer #1 (Remarks to the Author):

This manuscript describes microlaser sensors based on FRET-based lasing interaction between donor gain dyes present in a microsphere cavity and acceptor dyes in a sensing region. The “center wavelength (CWL)” of emission spectra from the donor and acceptor molecules is changed as a function of the concentration of the acceptor dye, which serves as the basis for sensing. This proof of concept is demonstrated inside cells in which both the donor-coated microspheres and acceptor molecules are present.

The observation of enhanced sensitivity by FRET is really interesting. Nonetheless, I have some basic questions from a physics point of view.

From the perspective of applications, however, it seems the sensitivity at the level of 10 micro-Molar is at least 2-3 orders of magnitude short of what's needed practically.

Here are specific comments.

1. The major finding of this work is the 10^4 enhancement of sensitivity compared to non-FRET lasing. I think its mechanism should be explained in more detail with rigor.
2. The key parameter is CWL. How is this determined? It seems that the emission spectra consist of many lasing peaks. Does the CWL refer to the wavelength of a peak with the maximum height, or a numerical mean of the spectral envelope?
3. If the former, I guess CWL would be a discrete, rather than continuous, function of concentration. If so, this would be a major issue for sensing and needs to be addressed in more detail.
4. Can CWL be determined unambiguously from a single measurement (without tracking the shift of peaks)? How reliable and repeatable is this measurement? The relative height of lasing peaks may fluctuate depending on the pumping condition.
5. The opposite sensitivity to acceptor concentration is interesting. If the definition of CWL was the mean wavelength of lasing peaks, the shift of gain-peak wavelength as a function of cavity loss makes sense. However, if the CWL refers to a lasing peak, it involves a shift of cavity resonance, and it is not clear whether the gain shift explains the negative direction of CWL shift for DG lasing.
6. I'd like to see CWL plots also at lower concentrations, e.g. 0 to 1 μM , a range more relevant to intracellular applications.
7. The concept proposal in Figure 6 is nice but seems too far-fetched without a feasibility analysis or demonstration. How many donor-acceptor linkers are needed, and is the sensitivity to target DNA and protein practical?
8. It is expected that the absolute value of CWL and sensitivity to R6G concentration depend on the initial concentration of DG. This has an implication to the accuracy of sensing. How much is this dependence?
9. What's the value of mode volume (referring to equation (1) or (2))? It is not obvious why the acceptor laser mode volume is 10^4 smaller than the non-FRET donor laser mode volume. A quantitative analysis for the observed sensitivity enhancement is needed.
10. I would recommend showing the entire output spectra in Fig. 3a and Fig. 4a.
11. Define parameters, such as α_{ex} in equation (1) and n_s and n_m in equation (2).

Reviewer #2 (Remarks to the Author):

The paper reported a microsphere laser based intracellular sensor that relies on FRET. The proposed idea is:

1. Microsphere is doped with donor dyes (in the current work, DG is used)
2. Immerse the microsphere in the acceptor solution (in the current work, R6G is used)
3. Excite the donor, and both donor and acceptor both lase.
4. The acceptor lasing is caused by the resonance energy transfer from the donor and the second cavity formed due to the FRET effect.
5. The lasing wavelength shift is proportional to the acceptor concentration. Therefore, the acceptor lasing wavelength can be used to estimate the acceptor concentration.

The main part of the paper describes the FRET laser from the dual cavity. In the end, as an application the microsphere is internalized by a cell filled with R6G in cytoplasm.

This is an interesting work and the phenomenon (such as the second cavity formed by the FRET effect) is something new. However, there are a number of flaws (or unanswered questions) that prevents me from recommending this paper.

1. Technical part

Major

(A) Sensing mechanism is not well articulated

(B) How is the second cavity formed? The authors attribute the second cavity (or virtual cavity) is formed by the FRET effect. The physics behind this effect is very vague and not justifiable.

(C) The time-resolved lifetime measurement is used to justify that there is FRET between DG and R6G. But this method is highly questionable. Fluorescence comes main from the microsphere body. The fluorescence contributed by the DG molecules near the sphere surface is very small. FRET occurs only between the DG molecules near the sphere surface and the R6G molecules with a few nanometers away from the sphere surface. I am surprised to see that the lifetime change in only a small fraction of DG molecules near the sphere surface can cause a huge lifetime change in the entire DG molecules (i.e., the DG molecules in the sphere body, plus DG molecules near or on the sphere surface).

Minor

Figure S3. I am surprised that no lasing was observed when 15 μ m sodalime glass microsphere was embedded in the 50 μ M R6G solution. R6G lasing has been observed many times with ring resonators (microsphere based, capillary based, etc.).

2. Application

The paper shows the FRET lasing when a microsphere is internalized by a cell filled with R6G. This application is less interesting and less practical. The best way is to demonstrate an actual sensing such as DNA hybridization on the microsphere surface or protein binding on the sphere surface.

Reviewer #3 (Remarks to the Author):

This is a very interesting manuscript that reports on using wgm FRET lasing with DG donor dye molecules doped into a polystyrene wgm microspheres and R6G acceptor molecules at or near the wgm microsphere in solution, for detecting rhodamine 6G at μ M concentration levels, in situ and inside of single cells. The detection is achieved by monitoring the wavelength blue and red shifts of the DG and R6G laser lines, respectively. Convincing evidence is provided for the FRET mechanism generating the lasing spectra and the wavelength shifts which are used for the sensitive (μ M) detection of R6G with extrapolated sensitivity approx. few nM R6G, which is excellent for this

approach.

Please consider revising the manuscript according to the following comments:

In their introduction, the authors could also mention other techniques that already detect /image even single fluorescent dye molecules inside of single cells, and how one compares these existing techniques to this novel work on intracellular wgm sensing of R6G.

In the discussion, the authors could mention how detection on their platform can be made specific to biomolecules of interest such as proteins, dna etc.

“near-universal feature of these precision optical biosensors is that high light intensities” : I believe that this statement is incorrect for wgm sensors. The coupling of a high laser light intensity into (passive) wgm sensors does not directly result in any improvements of the molecule detection sensitivity. The wgm interferometric signal of a passive wgm sensor does not directly increase with increasing the laser power coupled to the wgm. Hence, the authors may wish to rephrase their statement that the precise molecule sensing with wgm depends on high light intensities. The reader should not be confused by the local enhancements of near fields, which have been used in wgm sensing, with an incorrect claim that an overall high light intensity is required for the ultra-sensitive wgm sensing of molecules.

I suggest that the authors include (more) references for the observation of photo damage in passive wgm biomolecular sensing.

The authors may also like to compare the peak power densities encountered in the wgm evanescent field used for (fluorescent-free) molecule sensing to the peak power densities encountered in the focus of their pump beam that excites the active wgm microsphere cavities. The authors could then also compare all of these power density levels to the levels that are known to cause photo damage to dye molecules in (single-molecule) microscopy. I also note that their technique does rely on excitation of dye molecules which are naturally more prone to photo damage as compared to non-fluorescent molecules that are detected on a label free, passive wgm sensor.

Overall, I do not understand why the authors motivate their nice work with a problem of photodamage in wgm sensing. In my opinion, the main motivation for biosensing with active cavities is the in vivo sensing applications that are made possible by the free space excitation with the pump light. In my opinion, photo damage can only be made worse with the use of active cavities because of the requirement of pulsed laser excitation, dye molecules, and comparatively high peak power densities.

“As the polystyrene microsphere is negatively charged”: can the authors elaborate and provide experimental evidence for this negative charge? What is the buffer pH, pK of the chemical groups that carry the negative charge, etc.?

“Here, we demonstrate that incorporating Förster resonance energy transfer (FRET) to a WGM microcavity platform can greatly increase the sensitivity” I would ask the authors to be more specific about which version of the wgm sensor they talk about. This statement is generally speaking incorrect, I believe.

It is indeed very interesting in wgm FRET lasing sensing that increasing the concentration of acceptor molecules at μM concentration levels (where I assume any adsorption of the molecule to the sensor is always at saturation) results in blue shift of the donor lasing lines and a red shift for acceptor lasing lines. This makes for a very interesting and novel detection mechanism. In their control experiments, how do the authors know that the non fluorescent R6GH molecule adsorbs in the same way to the microsphere surface as R6G?

Why have the authors not varied the microsphere surface charge to test their hypothesis that surface adsorption of R6G to the native charged microsphere is required for FRET lasing to occur?

Have the authors considered that the slight heating of the microsphere may play a role in the shifts of the donor and acceptor lasing lines? How does the shift depend on the laser power/intensity?

The R6G redshift could be due to increase of refractive index in the higher R6G solution concentration? Can this be ruled out?

“increased losses lead to a frequency shift towards shorter wavelengths”... perhaps the authors might want to elucidate this mechanism a bit more... I am not sure if the interdisciplinary readership of this journal can follow their argument.

Can the authors comments on the fundamental problem of using active wgm cavities for ultra-sensitive detection of gain molecules as this will always required a minimal concentration of gain molecules? Can one estimate this minimal concentration of dye molecules that is required to build a FRET laser which I assume also depends on the Q factor?

Response letter (NCOMMS-20-41451A)

REVIEWER COMMENTS

Reviewer #1 (Remarks to the Author):

This manuscript describes microlaser sensors based on FRET-based lasing interaction between donor gain dyes present in a microsphere cavity and acceptor dyes in a sensing region. The “center wavelength (CWL)” of emission spectra from the donor and acceptor molecules is changed as a function of the concentration of the acceptor dye, which serves as the basis for sensing. This proof of concept is demonstrated inside cells in which both the donor-coated microspheres and acceptor molecules are present.

The observation of enhanced sensitivity by FRET is really interesting. Nonetheless, I have some basic questions from a physics point of view.

From the perspective of applications, however, it seems the sensitivity at the level of 10 micro-Molar is at least 2-3 orders of magnitude short of what’s needed practically.

Thanks to the reviewer for raising the important point. In the original manuscript, we only showed the data of R6G $\geq 1 \mu\text{M}$. To address the reviewer’s concern, we have included concentrations within range of 0-1000 nM, which is the common concentration range for intracellular molecules. With the inclusion of lower concentration data we were able to show that the detection limit is at least in the pM level, which is sufficient for subcellular applications. We have included the data in the revised MS, please refer to page 11, last paragraph.

In addition, we would say that what we achieved with the present setup presumably is not the limit of sensitivity. The sensitivity could be further enhanced via optimizing conditions with respect to pulse length, repetition rate, diameter & refractive index of the microsphere and so forth, which is an important section of work we are currently carrying on.

Here are specific comments.

1. The major finding of this work is the 10^4 enhancement of sensitivity compared to non-FRET lasing. I think its mechanism should be explained in more detail with rigor.

Thanks for the constructive comment. Following the reviewer’s advice, we have done some theoretical study and mechanism exploration of the FRET-WGM system; details are as follows.

We have also included the content in the revised supplementary information (SI section 3).

The molecular processes between donor and acceptor in FRET are illustrated by the Jablonski diagram ^[1] (figure 1 below). When donor molecules in the microcavity are pumped by a laser, photons transit to S_{1d} from S_{0d} . Since all conditions meet for FRET to occur, the excited DG molecules will transfer part of the energy to the R6G molecules, which will result in R6G emission, i.e. energy transition occurs from S_{1a} to S_{0a} .

Fig.1 The energy transfer chart of FRET

In the FRET-WGM system, we use coupled differential equations to explore the insight of the system. The coupled differential equations used to describe the dynamics of excite-state molecule density and photon density of the FRET donor-acceptor pairs are as follows:

$$\frac{dn_d(t)}{dt} = I_p(t)[N_d - n_d(t)]\sigma_{pd} + \frac{cq_d(t)}{n_1}\sigma_{add}[N_d - n_d(t)] - \frac{cq_d(t)}{n_1}\sigma_{edd}n_d(t) - k_F n_d(t) - \frac{n_d(t)}{\tau_d} \quad (1)$$

$$\frac{dq_d(t)}{dt} = \frac{c}{n_1 V}\sigma_{edd}n_d(t) + \frac{cq_d(t)}{n_1}\sigma_{edd}n_d(t) - \frac{cq_d(t)}{n_1}\sigma_{add}[N_d - n_d(t)] - \frac{q_d(t)}{\tau_{ed}} + \frac{cq_d(t)}{n_2}\sigma_{ead}n_a(t) - \frac{cq_d(t)}{n_1}\sigma_{aad}[N_a - n_a(t)] \quad (2)$$

$$\frac{dn_a(t)}{dt} = I_p(t)[N_a - n_a(t)]\sigma_{pa} + \frac{cq_a(t)}{n_2}\sigma_{aaa}[N_a - n_a(t)] - \frac{cq_a(t)}{n_2}\sigma_{eaa}n_a(t) + k_F n_d(t) - \frac{n_a(t)}{\tau_a} + \frac{cq_d(t)}{n_1}\sigma_{aad}[N_a - n_a(t)] - \frac{cq_d(t)}{n_2}\sigma_{ead}n_a(t) \quad (3)$$

$$\frac{dq_a(t)}{dt} = \frac{c}{n_2 V}\sigma_{eaa}n_a(t) + \frac{cq_a(t)}{n_2}\sigma_{eaa}n_a(t) - \frac{cq_a(t)}{n_2}\sigma_{aaa}[N_a - n_a(t)] - \frac{q_a(t)}{\tau_{ea}} \quad (4)$$

In which, $n_d(t)$, $n_a(t)$, $q_d(t)$, and $q_a(t)$ represent molecular densities of donor and acceptor dye in the excited state, and emitted photon densities of donor and acceptor dye, respectively. N_d and N_e are the total number of donor and acceptor dye molecules, respectively. σ_{pd} and σ_{pa} , respectively, describe the absorption cross sections of the donor and acceptor at the excitation wavelength. σ_{edd} , σ_{add} , σ_{ead} and σ_{aad} are the donor emission, donor absorption, acceptor emission and acceptor absorption cross sections, respectively, at the donor lasing wavelength. σ_{aaa} and σ_{eaa} are the acceptor emission and acceptor absorption cross sections, respectively, at the acceptor dye lasing wavelength. $I_p(t)$ is the time-dependent pump intensity and k_F is the FRET rate. n_1 and n_2 respectively, represent the resonant cavity and surrounding medium refractive index. V is the mode volume, and τ_d , τ_a , τ_{ed} and τ_{ea} , respectively, denote the fluorescence lifetimes of the donor and acceptor, and the cavity decay times at the excitation wavelength of donor and acceptor.

In Equations (1) and (3), the first terms on the right-hand side indicate the number of donors or

acceptors directly excited by a laser pump on the ground state energy level. The second term designates at the donor excitation wavelength, the number of donors or acceptors at the ground state level that are excited by absorbing the photon energy emitted by the donor; were derived based on the spontaneous emission in laser mode ^[2-4]. The third terms represent that at the donor or acceptor excitation wavelength, the reduced numerical densities of the excited state donor or acceptor when the donor or acceptor of the excited state transitions to the ground state energy level through stimulated radiation. The fourth terms are the most important ones, representing the decrease in the numerical densities of the excited state donor or the increase in the numerical densities of excited state acceptor after fluorescence resonance energy transfer. The fifth terms indicate that the numerical densities of donor or acceptor in the excited state are reduced due to the fluorescence lifetime. For equation (3), the last two terms represent that, at the donor excitation wavelength, the increased numerical densities of the excited state acceptor when the ground state acceptor transitions to the excited state by absorbing photon energy emitted by the donor and the decreased numerical densities of the excited state acceptor when the excited state acceptor transitions to the ground state energy level through stimulated radiation, respectively.

In Equations (2) and (4), the first terms on the right-hand side are derived based on the spontaneous emission of the excited donor or acceptor molecules ^[2-4]. The second terms indicate that at the corresponding excitation wavelength, the numerical densities of photons emitted when the excited state donor or acceptor transitions to the ground state energy level through stimulated radiation. The third terms designate the reduced numerical densities of photons, when the ground state donor or acceptor transitions to the excited state energy level by absorbing the photon energy emitted by the donor or acceptor. The fourth terms represent the numerical densities of photons emitted by the donor or acceptor is reduced due to the cavity decay times. For equation (2), the last two terms indicate, at the donor excitation wavelength, the increased numerical densities of photons emitted when the excited state acceptor transitions to the ground state energy level through stimulated radiation and the reduced numerical densities of photons, when the donor transitions to the excited state energy level by absorbing the photon energy emitted by the donor.

In the FRET-WGM system described in our MS, the FRET efficiency is high. We obtained FRET efficiency as high as 58% ($E_{\text{FRET}}=(\tau_{\text{D}}-\tau_{\text{DA}})/\tau_{\text{DA}}$)^[5] (please refer to revised MS, page 9 first paragraph), which is much higher than reported data in the past^[6-9]. The great FRET efficiency implies that energy transferred to R6G, the acceptor molecule, via FRET itself is sufficient to achieve population inversion, that is to say, FRET dominates the process for R6G lasing. In this case, refractive index change is not the dominating factor.

The energy loss of DG molecules marks a blue shift of the resonance wavelength; the shift is dependent on the surrounding R6G concentration, which is proven by the increased lasing threshold of DG. Meanwhile, the resonance wavelength of acceptor R6G molecules marks a red-shift due to receiving energy. The sensitivity of the microcavity detection is determined by the wavelength gap ($\Delta\lambda$) between donor and acceptor emission, $\Delta\lambda=\lambda_{\text{a}}-\lambda_{\text{d}}$.

While in the case of pure WGM (non-FRET-WGM) system (we are talking about active mode

WGM here), as documented in many publications, the shift of the resonance wavelength is mainly dependent on the refractive index. For R6GH solution, the change of refractive index from one concentration to another is not very profound, see the table below, which corresponds to a small $\Delta\lambda$ ($\lambda-\lambda_0$). Although refractive index change of R6G is equally minor to R6GH (see table below), the FRET-WGM sensing performance is not determined by refractive index change. This explains the big difference of sensitivity between pure active WGM and FRET-WGM.

Table 1. Refractive index data of R6G and R6GH measured by an Abbe refractometer.

R6G concentration	refractive index (n)	Δn (n-n0)
0 (H2O)	1.332	
10nM	1.3351	0.0029
100nM	1.3354	0.0032
10uM	1.3361	0.0039
R6GH concentration		
0 (90%H2O+10%EtOH)	1.3346	
10nM	1.3367	0.0021
100nM	1.3375	0.0029
10uM	1.3382	0.0036

note: R6G and R6GH are dissolved in H2O and 90%H2O+10%EtOH, respectively.

References:

- [1]. Jablonski, A. Efficiency of Anti-Stokes Fluorescence in Dyes. *Nature* 131, 839–840 (1933).
- [2]. Hebling J, Seres J, Bor Z, et al. Dye laser pulse shortening and stabilization by Q-switching[J]. *Optical & Quantum Electronics*, 1990, 22(4):375-384.
- [3]. Atkinson J, Pace F. The spectral linewidth of a flashlamp-pumped dye laser[J]. *IEEE Journal of Quantum Electronics*, 1973, 9(6):569-574.
- [4]. Aas M, Chen Q, Jonas A, et al. Optofluidic FRET Lasers and Their Applications in Novel Photonic Devices and Biochemical Sensing[J]. *IEEE Journal of Selected Topics in Quantum Electronics*, 2015, 22(4):188-202.
- [5]. Shopova, S.I., Cupps, J.M., Zhang, P., Henderson, E.P. & Fan, X.D. Opto-fluidic ring resonator lasers based on highly efficient resonant energy transfer. *Optics Express*, 15(20), 12735-42 (2007).
- [6]. Chakraborty U, Maiti P, Singha T, et al. Effect of montmorillonite clay on the fluorescence resonance energy transfer between two cationic dyes Acridine Orange and Rhodamine B in solution[J]. *Materials today: proceedings*, 2020.
- [7]. Chakraborty S, Arshad Hussain S. Fluorescence resonance energy transfer (FRET) between acriflavine and CdTe quantum dot[J]. *Materials Today: Proceedings*, 2020.
- [8]. Zambrana-Puyalto X, Ponzellini P, Nicolò Maccaferri, et al. Förster-Resonance Energy Transfer between Diffusing Molecules and a Functionalized Plasmonic Nanopore[J]. *Physical Review Applied*, 2020.
- [9]. Pramanik A, Biswas S, Sekhar Tiwary C, et al. Forster resonance energy transfer assisted white light generation and luminescence tuning in a colloidal graphene quantum dot-dye system[J].

2. The key parameter is CWL. How is this determined? It seems that the emission spectra consist of many lasing peaks. Does the CWL refer to the wavelength of a peak with the maximum height, or a numerical mean of the spectral envelope?

Thanks for the comments. In our original MS, CWL (central wavelength) is defined as the wavelength of the center peak in the group of WGM peaks. Most of the time, the height of the center peak is consistently the maximum, since the center peak is obtained under the best resonant condition, i.e. the minimum energy loss. We didn't observe much variation with respect to the relative height of the peaks upon changing the pumping condition. Only when we increase the pump energy to very high level, say, at which the lasing intensity is saturated on the threshold plot (figure 2 below, left panel), we observed the height of the central peak is not the maximum. In our experiments, we always keep the pumping energy below saturation. So we agree with the reviewer that the peak with maximum height is not always equal to center peak under extreme pumping conditions. Although the peak with maximum height is a discrete function, the CWL, as we observed throughout the experiments, only shifts with change of concentration, and does not "jump" from one maximum to the next.

After receiving the reviewer's comment, we compared the mean of the spectral envelope (MSE) with the CWL. We calculated the MSE and plotted a concentration dependent MSE and CWL histogram, please see figure 2 below, right panel. It shows that that the measured wavelengths at different R6G concentration using either CWL or MSE show comparable numbers.

As the center peak can be easily identified without ambiguousness (please refer to Q4 for detailed descriptions), it is much more convenient to use CWL as a measure for the sensing system.

Figure 2. Left panel: lasing profile under various pumping conditions. Inset: threshold chart. Right panel: Concentration-dependent wavelength shift expressed using MSE or CWL.

3. If the former, I guess CWL would be a discrete, rather than continuous, function of concentration. If so, this would be a major issue for sensing and needs to be addressed in more detail.

Thanks for the comments. As we have discussed in the Q2, the CWL and MSE are giving same results.

4. Can CWL be determined unambiguously from a single measurement (without tracking the shift

of peaks)? How reliable and repeatable is this measurement? The relatively height of lasing peaks may fluctuate depending on the pumping condition.

Thanks for the comments. We think the CWL can be determined unambiguously from a single measurement, as long as we don't use saturated pumping energy. We measured the lasing peaks of nine randomly selected microspheres and plotted the spectra shown in figure 3 below. We can see the central peaks and the CWLs are consistently located at the same position. In addition, the reproducibility of the inter-experimental data is very high. The data presented in the MS were average of 5 repeated experiments. Since Dr Marion Lang (one of the first authors) initiated the work, there have been four authors taking over the experimental work over a time-span of five years. They all obtained same results with respect to the wavelength shift. So we are confident with the results. We agree with the reviewer that the peak with maximum height may fluctuate depending on the pumping condition, but as discussed in our answer to Q2, as long as we keep the pumping energy below the saturation level, we can get clear-cut data.

Figure 3: Laser emission of randomly selected DG microspheres.

5. The opposite sensitivity to acceptor concentration is interesting. If the definition of CWL was the mean wavelength of lasing peaks, the shift of gain-peak wavelength as a function of cavity loss makes sense. However, if the CWL refers to a lasing peak, it involves a shift of cavity resonance, and it is not clear whether the gain shift explains the negative direction of CWL shift for DG lasing. Thanks for the comments. As discussed in the former questions, the CWL and MSE are giving comparable results. And CWL can be easily identified without ambiguousness. Therefore, for convenience, we use CWL as a measure for studying the sensing performance.

6. I'd like to see CWL plots also at lower concentrations, e.g. 0 to 1 μ M, a range more relevant to intracellular applications.

Thanks for the constructive comments. Following the reviewer's advice, we carried out more experiments to measure the CWLs of DG and R6G at lower R6G concentrations ranging from 0-1 μ M (0, 10nM, 50nM, 75nM, 100nM, 500nM). The updated plot is shown in figure 4 below (left panel). Meanwhile, we also re-performed the control experiment using the R6G analog, R6GH,

again including lower concentration data, please see figure 4 right panel. Using the same detection limit, *i.e.* the linewidth $\delta\lambda$, we calculated the limits of detection (LOD) of the two configurations. The LODs are 61.1pM for FRET-WGM and 632.5nM for non-FRET-WGM, respectively. The $\text{LOD}_{\text{FRET-WGM}}$ is still 10^4 times lower than that of non-FRET-WGM system. The data were included in our revised MS. Please refer to Fig 3, Fig 4c,d and page 11-13 in the revised MS.

Here we arbitrarily defined the limit of detection as the line width $\delta\lambda$. In reality, the smallest detectable wavelength shift is usually limited by the resolution of the spectrometer, which can be much smaller than the line width $\delta\lambda$. In our case, the resolution of our spectrometer is 0.04nm. If we use 0.04nm as the limit, the calculated LOD for FRET-WGM is 20.3pM, and 313nM for pure WGM, which is 1.56×10^4 times difference.

We would say the LOD can be further enhanced via optimizing the system. Parameters such as pulse length, refractive index of the microsphere, doping quantity of fluorescent molecule, and so forth, can be fine-tuned to reach a maximum enhancement. That will be a comprehensive job. We hope to further explore that in the near future.

Figure 4. $\Delta\lambda$ ($\lambda_{\text{R6G}} - \lambda_{\text{DG}}$)-concentration curve of the FRET-WGM system (left panel) and $\Delta\lambda$ ($\lambda - \lambda_0$)-concentration curve of pure WGM system (right panel).

7. The concept proposal in Figure 6 is nice but seems too far-fetched without a feasibility analysis or demonstration. How many donor-acceptor linkers are needed, and is the sensitivity to target DNA and protein practical?

Thanks for the comments. To make the figure more relevant, we have added more text to explain the figure, please see the legend of figure 6 in the revised MS. Practically, we have proposed an experiments using FRET-WGM for microRNA (miRNA) sensing. The experiment is currently being conducted in the lab. Since this work is ongoing at the moment, we only can outline the experiment briefly:

Intracellular microRNA (miRNA) measurement at single cell resolution. Briefly: modifying the microsphere with a single strand DNA (ssDNA) sequence, namely recognizing sequence, which is complementary to the target miRNA. Then, hybridization of a short reporter DNA sequence labelled with R6G. The proposed configuration allows R6G molecules to be sterically very close to the microsphere surface, enabling FRET-WGM lasing to occur. In the presence of the target miRNA

molecule, the reporter DNA dissociates from the hybrid and results in a wavelength shift of lasing spectra.

Introducing specific recognizing elements on the microsphere surface not only can eliminate false results caused by non-specific binding allowing targeted sensing at single cell level, they also can enrich the target molecules.

The amount of donor-acceptor linkers can be estimated roughly. Since we know the R6G concentration that can cause a detectable wavelength shift, say 100nM, we could modify the surface with 100nM recognizing sequence. After hybridization, there would be 100nM R6G molecules located on the microsphere, since the two strands of DNA hybridise at a 1:1 ratio. This way, we can be sure that there are enough R6G molecules for sensing the intracellular change of target molecules. Of course, this is just a blueprint, in practice; various experiments need to be conducted to optimize the system.

As documented in many publications, a distinct advantage of FRET measurement is its dynamic feature. Via integration of FRET and active WGM, real-time observation of the dynamic processes in a cell becomes feasible.

8. It is expected that the absolute value of CWL and sensitivity to R6G concentration depend on the initial concentration of DG. This has an implication to the accuracy of sensing. How much is this dependence?

Thanks for the constructive comment. We agree with the reviewer. The CWL value would be related to the gain molecule concentration doped in the microsphere. To confirm this, we ordered higher-quantity DG doped microsphere (2% DG vs 1% DG). Since the supplier is located in the States, plus this order is a custom job, it took very long time to come (and very expensive as well). We have just finished the experiments recently and the data are shown below (figure 5). All parameters of the newly fabricated microsphere are exactly same as the previous one except for higher DG doping.

Increasing DG concentration in the microsphere to 2.0% w/w, similar lasing performance was observed (Figure 5a, b and c). The CWL of DG in H₂O is 539nm, red-shifted by 5 nm compared with 1% DG microsphere, due to higher gain molecule quantity in the resonator. Embedding the microsphere in various concentrations of R6G and pumping with 473nm laser, similarly, we observed dose-dependent blue- and red-shift of DG and R6G lasing CWLs, respectively; Figure 5d, e. At the same R6G concentration, the wavelength distance between DG and R6G ($\Delta\lambda = \lambda_{R6G} - \lambda_{DG}$), Figure 5f, is larger with 2% DG-doping than that of 1% DG-doping, as shown in table 2. The LOD_{FRET-WGM} of the 2% DG microcavity sensor, which is calculated to be 2.2pM, is much lower than that of 1% counterpart (61.1pM). We also compared the sensing performance of FRET-WGM and non-FRET-WGM using a 2% DG microsphere. The microsphere was ingrained in R6GH solution and the lasing action of DG was evaluated (Figure 5g, h). The LOD without FRET was 76.5nM, which is 3.5×10^4 times less sensitive than the FRET-WGM system. The data, on the one hand, further confirmed the 1% DG microsphere results; more importantly,

also provide a valid path to further optimize the sensitivity of the FRET-WGM platform.

Figure 5. FRET-WGM sensing performance using a 2% DG doped microcavity. **a.** DG emission when a 2% DG doped microsphere was embedded in H₂O and pumped with a 473nm laser. **b.** FSR and line width. **c.** Lasing threshold. The lasing threshold is lower than that of a 1% DG doped microsphere. **d.** Dose-dependent lasing spectra of DG and R6G. Only CWs are shown. **e.** Dose-dependent wavelength curve of DG and R6G. **f.** dose-dependent $\Delta\lambda$ ($\lambda_{R6G}-\lambda_{DG}$) curve. **g.** DG emission when a 2% DG doped microsphere was embedded in R6GH solution. Only center peaks are shown on the picture. **h.** dose-dependent $\Delta\lambda$ ($\lambda_{DG-in-R6GH}-\lambda_{DG-in-solven}$) curve.

Table 2. $\Delta\lambda$ ($\lambda_{R6G}-\lambda_{DG}$) values at same R6G concentration with different quantity of DG-doping

R6G Conc.	10nM	50nM	100nM	500nM	1 μ M	5 μ M	10 μ M	25 μ M	50 μ M	75 μ M	100 μ M	200 μ M	300 μ M	400 μ M	500 μ M
$\Delta\lambda$ (nm) (1% DG)	6.15	8.93	9.63	10.59	12.81	20.81	32.53	36.54	42.29	45.52	47.50	49.49	50.75	53.36	55.23
$\Delta\lambda$ (nm) (2% DG)	3.56	8.68	11.93	17.17	20.85	26.93	35.61	41.44	45.23	47.71	49.92	53.35	55.41	56.75	57.07

We have included the data in the revised MS, please refer to page 15, the last paragraph, page 16, and figure S10 in SI.

9. What's the value of mode volume (referring to equation (1) or (2))? It is not obvious why the acceptor laser mode volume is 10^4 smaller than the non-FRET donor laser mode volume. A quantitative analysis for the observed sensitivity enhancement is needed.

Thanks for the comments. Equations (1) and (2) are referred here to demonstrate that the LOD can be further enhanced via optimizing the mode volume. In the MS, we have not investigated the mode volume of FRET-WGM and non-FRET WGM. We are sorry that maybe our wording led to some ambiguity.

10. I would recommend showing the entire output spectra in Fig. 3a and Fig. 4a.

Thanks for the comment. In the revised MS, the entire output spectra were shown in the supplementary materials, please refer to figure S7.

11. Define parameters, such as α_{ex} in equation (1) and n_s and n_m in equation (2).

Thanks for the comments. α_{ex} is the polarizability of the particles (molecules) bound to the microcavity, n_s and n_m are the refractive indices of the sphere and exterior medium, respectively. We have fully defined the parameters in equation (1) and (2) in the revised MS.

Finally, we would like to thank the referee again for taking time to review our manuscript.

Reviewer #2 (Remarks to the Author):

The paper reported a microsphere laser based intracellular sensor that relies on FRET. The proposed idea is:

1. Microsphere is doped with donor dyes (in the current work, DG is used)
2. Immerse the microsphere in the acceptor solution (in the current work, R6G is used)
3. Excite the donor, and both donor and acceptor both lase.
4. The acceptor lasing is caused by the resonance energy transfer from the donor and the second cavity formed due to the FRET effect.
5. The lasing wavelength shift is proportional to the acceptor concentration. Therefore, the acceptor lasing wavelength can be used to estimate the acceptor concentration.

The main part of the paper describes the FRET laser from the dual cavity. In the end, as an application the microsphere is internalized by a cell filled with R6G in cytoplasm.

This is an interesting work and the phenomenon (such as the second cavity formed by the FRET effect) is something new. However, there are a number of flaws (or unanswered questions) that prevents me from recommending this paper.

1. Technical part

Major

(A) Sensing mechanism is not well articulated

Thanks to the reviewer for raising this important point. We are sorry for not describing the sensing mechanism clearly in the original MS. In response to reviewer's comment, we have added mechanism exploration of the FRET-WGM system. Please refer to the following text and SI section 3.

The molecular processes between donor and acceptor in FRET are illustrated by the Jablonski diagram ^[1] (figure 1 below). When donor molecules in the microcavity are pumped by laser, photons transit to S_{1d} from S_{0d} . When all conditions for FRET to occur are met, the excited DG molecules will transfer part of the energy to the R6G molecules, which will result in R6G emission, i.e. energy transition occurs from S_{1a} to S_{0a} .

Fig.1 The energy transfer chart of FRET.

We applied coupled differential equations to explore the insight of FRET-WGM system. The coupled differential equations used to describe the dynamics of excite-state molecule density and photon density of the FRET donor-acceptor pairs are as follows:

$$\frac{dn_d(t)}{dt} = I_p(t)[N_d - n_d(t)]\sigma_{pd} + \frac{cq_d(t)}{n_1}\sigma_{add}[N_d - n_d(t)] - \frac{cq_d(t)}{n_1}\sigma_{edd}n_d(t) - k_F n_d(t) - \frac{n_d(t)}{\tau_d} \quad (1)$$

$$\begin{aligned} \frac{dq_d(t)}{dt} = & \frac{c}{n_1 V}\sigma_{edd}n_d(t) + \frac{cq_d(t)}{n_1}\sigma_{edd}n_d(t) - \frac{cq_d(t)}{n_1}\sigma_{add}[N_d - n_d(t)] - \frac{q_d(t)}{\tau_{ed}} + \frac{cq_d(t)}{n_2}\sigma_{ead}n_a(t) \\ & - \frac{cq_d(t)}{n_1}\sigma_{aad}[N_a - n_a(t)] \end{aligned} \quad (2)$$

$$\begin{aligned} \frac{dn_a(t)}{dt} = & I_p(t)[N_a - n_a(t)]\sigma_{pa} + \frac{cq_a(t)}{n_2}\sigma_{aaa}[N_a - n_a(t)] - \frac{cq_a(t)}{n_2}\sigma_{eaa}n_a(t) + k_F n_d(t) - \frac{n_a(t)}{\tau_a} \\ & + \frac{cq_d(t)}{n_1}\sigma_{aad}[N_a - n_a(t)] - \frac{cq_d(t)}{n_2}\sigma_{ead}n_a(t) \end{aligned} \quad (3)$$

$$\frac{dq_a(t)}{dt} = \frac{c}{n_2 V}\sigma_{eaa}n_a(t) + \frac{cq_a(t)}{n_2}\sigma_{eaa}n_a(t) - \frac{cq_a(t)}{n_2}\sigma_{aaa}[N_a - n_a(t)] - \frac{q_a(t)}{\tau_{ea}} \quad (4)$$

In which, $n_d(t)$, $n_a(t)$, $q_d(t)$, and $q_a(t)$ represent molecular densities of donor and acceptor dye in the excited state, and emitted photon densities of donor and acceptor dye, respectively. N_d and N_e are the total number of donor and acceptor dye molecules, respectively. σ_{pd} and σ_{pa} , respectively, describe the absorption cross sections of the donor and acceptor at the excitation wavelength. σ_{edd} , σ_{add} , σ_{ead} and σ_{aad} are the donor emission, donor absorption, acceptor emission and acceptor absorption cross sections, respectively, at the donor lasing wavelength. σ_{aaa} and σ_{eaa} are the acceptor emission and acceptor absorption cross sections, respectively, at the acceptor dye lasing wavelength. $I_p(t)$ is the time-dependent pump intensity and k_F is the FRET rate. n_1 and n_2 respectively, represent the resonant cavity and surrounding medium refractive index. V is the mode volume, and τ_d , τ_a , τ_{ed} and τ_{ea} , respectively, denote the fluorescence lifetimes of the donor and acceptor, and the cavity decay times at the excitation wavelength of donor and acceptor.

In Equations (1) and (3), the first terms on the right-hand side indicates the number of donors or acceptors directly excited by a laser pump on the ground state energy level. The second term designates at the donor excitation wavelength, the number of donors or acceptors at the ground state level that are excited by absorbing the photon energy emitted by the donor; were derived based on the spontaneous emission in laser mode^[2-4]. The third terms represent that at the donor or acceptor excitation wavelength, the reduced numerical densities of the excited state donor or acceptor when the donor or acceptor of the excited state transitions to the ground state energy level through stimulated radiation. The fourth terms are the most critical one, representing the decrease in the numerical densities of excited state donor or the increase in the numerical densities of excited state acceptor after fluorescence resonance energy transfer. The fifth terms indicate the numerical densities of donor or acceptor in the excited state is reduced due to the fluorescence lifetime. For equation (3), the last two terms represent that, at the donor excitation wavelength, the increased numerical densities of the excited state acceptor when the ground state acceptor transitions to the excited state by absorbing photon energy emitted by the donor and the decreased numerical densities of the excited state acceptor when the excited state acceptor

transitions to the ground state energy level through stimulated radiation, respectively.

In Equations (2) and (4), the first terms on the right-hand side were derived based on the spontaneous emission of the excited donor or acceptor molecules [2-4]. The second terms indicate that at the corresponding excitation wavelength, the numerical densities of photons emitted when the excited state donor or acceptor transitions to the ground state energy level through stimulated radiation. The third terms designate the reduced numerical densities of photons, when the ground state donor or acceptor transitions to the excited state energy level by absorbing the photon energy emitted by the donor or acceptor. The fourth terms represent the numerical densities of photons emitted by the donor or acceptor is reduced due to the cavity decay times. For equation (2), the last two terms indicate, at the donor excitation wavelength, the increased numerical densities of photons when the excited state acceptor transitions to the ground state energy level through stimulated radiation and the reduced numerical densities of photons, when the donor transitions to the excited state energy level by absorbing the photon energy emitted by the donor.

For the FRET-WGM system, the FRET efficiency is high. We obtained FRET efficiency as high as 58% ($E_{\text{FRET}}=(\tau_{\text{D}}-\tau_{\text{DA}})/\tau_{\text{DA}}$)^[5], which is much higher than reported data in the past^[6-9]. The great FRET efficiency implies that energy transferred to R6G, the acceptor molecule, via FRET itself is sufficient to achieve population inversion, that is to say, FRET dominates the process for R6G lasing. In this case, refractive index change is not the dominating factor.

The energy loss of DG molecules marks a blue shift of the resonance wavelength; the shift is dependent on the surrounding R6G concentration, which is proved by the increased lasing threshold of DG. Meanwhile, the resonance wavelength of acceptor R6G molecules marks a red-shift due to receiving energy. The sensitivity of microcavity detection is determined by the wavelength gap ($\Delta\lambda$) between donor and acceptor emission, $\Delta\lambda=\lambda_a-\lambda_d$.

In the case of pure a WGM (i.e.. non-FRET-WGM) system (we are describing an active mode WGM in our MS), as documented in many publications, the shift of the resonance wavelength is mainly dependent on the refractive index. For R6GH solution, the change of refractive index from one concentration to another is not very large, see table below, corresponding to a small shift in wavelength $\Delta\lambda$ ($\lambda-\lambda_0$). Although the refractive index change of R6G is comparable to the refractive index change of R6GH (see table below), we observe in fact a much larger shift than can be expected from the refractive index change. In conclusion, the FRET-WGM sensing performance is not determined by the refractive index change. This explains the big difference of sensitivity between pure active WGM and FRET-WGM.

Table 1. Refractive index data of R6G and R6GH measured by an Abbe refractometer.

R6G concentration	refractive index (n)	Δn (n-n0)
0 (H2O)	1.332	
10nM	1.3351	0.0029
100nM	1.3354	0.0032
10uM	1.3361	0.0039

R6GH concentration	refractive index (n)	Δn (n-n0)
0 (90%H2O+10%EtOH)	1.3346	
10nM	1.3367	0.0021
100nM	1.3375	0.0029
10uM	1.3382	0.0036

note: R6G and R6GH are dissolved in H2O and 90%H2O+10%EtOH, respectively.

References:

- [1]. Jablonski, A. Efficiency of Anti-Stokes Fluorescence in Dyes. *Nature* 131, 839–840 (1933).
- [2]. Hebling J, Seres J, Bor Z, et al. Dye laser pulse shortening and stabilization by Q-switching[J]. *Optical & Quantum Electronics*, 1990, 22(4):375-384.
- [3]. Atkinson J, Pace F. The spectral linewidth of a flashlamp-pumped dye laser[J]. *IEEE Journal of Quantum Electronics*, 1973, 9(6):569-574.
- [4]. Aas M, Chen Q, Jonas A, et al. Optofluidic FRET Lasers and Their Applications in Novel Photonic Devices and Biochemical Sensing[J]. *IEEE Journal of Selected Topics in Quantum Electronics*, 2015, 22(4):188-202.
- [5]. Shopova, S.I., Cupps, J.M., Zhang, P., Henderson, E.P. & Fan, X.D. Opto-fluidic ring resonator lasers based on highly efficient resonant energy transfer. *Optics Express*, 15(20), 12735-42 (2007).
- [6]. Chakraborty U, Maiti P, Singha T, et al. Effect of montmorillonite clay on the fluorescence resonance energy transfer between two cationic dyes Acridine Orange and Rhodamine B in solution[J]. *Materials today: proceedings*, 2020.
- [7]. Chakraborty S, Arshad Hussain S. Fluorescence resonance energy transfer (FRET) between acriflavine and CdTe quantum dot[J]. *Materials Today: Proceedings*, 2020.
- [8]. Zambrana-Puyalto X, Ponzellini P, Nicolò Maccaferri, et al. Förster-Resonance Energy Transfer between Diffusing Molecules and a Functionalized Plasmonic Nanopore[J]. *Physical Review Applied*, 2020.
- [9]. Pramanik A, Biswas S, Sekhar Tiwary C, et al. Forster resonance energy transfer assisted white light generation and luminescence tuning in a colloidal graphene quantum dot-dye system[J]. *Journal of Colloid and Interface Science*. 565, 326-336. 2020

(B) How is the second cavity formed? The authors attribute the second cavity (or virtual cavity) is formed by the FRET effect. The physics behind this effect is very vague and not justifiable.

Thanks for the comments. To address the reviewer's comment, we performed additional experiments. We suggest that the second cavity is formed via strong electrostatic adsorption between DG microsphere and R6G molecules. The charging properties of DG microsphere and R6G molecules were measured using a Zetasizer Nano ZS (Malvern Panalytical CO.), as shown in figure 2 below. The surface of the DG microspheres is negatively charged, while the R6G molecules are positively charged. Therefore, R6G molecules are readily adsorbed on the surface of the microsphere through hetero-electric attraction, forming a core-shell-like structure, as illustrated in figure 3 below.

Figure 2. Zeta potentials of DG microsphere and R6G molecules.

Figure 3. Schematic diagram of optical dual-cavity structure

The R6G shell is essentially a WGM microcavity. However, the R6G shell alone is not sufficient to support R6G lasing under existing conditions, as described in the MS using non-fluorescent sodalime microspheres in a control experiment. Only with the help of FRET, R6G molecules can lase at the present laser power. For the control experiment, a sodalime glass microsphere, with identical refractive index, charging property and diameter as the previously used DG-doped PS microspheres, was embedded in 50 μ M R6G solution. Pumping the microsphere using a 473nm laser under the same conditions with respect to R6G concentration and pumping energy as in our previous experiments, we only observed R6G fluorescence emission. (Here we would like to clarify that R6G lasing can be realized using 532nm pulsed laser pumping in our setup, however only under the condition that the R6G concentration is high enough (>5mM) and the pumping energy is elevated to >mJ level, as demonstrated in the supplementary information figure S3). In contrast, when the sodalime microsphere was replaced with DG-doped microsphere, pumping under even much lower energy conditions (nJ-level), we readily observed R6G lasing. Furthermore, via measuring the DG fluorescence lifetime doped in the microsphere in the absence/presence of R6G, we conclude that there is a highly efficient energy transfer between DG and R6G, which confirm that FRET is the predominant effect. The laser energy generated by DG molecules on the cavity surface is transferred to the R6G molecules through this non-radiative transition, and brings R6G molecules

in the excited state. With the help of the R6G shell WGM cavity formed at the outer surface of the DG microspheres, the R6G molecules are easily lasing.

(C) The time-resolved lifetime measurement is used to justify that there is FRET between DG and R6G. But this method is highly questionable. Fluorescence comes mainly from the microsphere body. The fluorescence contributed by the DG molecules near the sphere surface is very small. FRET occurs only between the DG molecules near the sphere surface and the R6G molecules with a few nanometers away from the sphere surface. I am surprised to see that the lifetime change in only a small fraction of DG molecules near the sphere surface can cause a huge lifetime change in the entire DG molecules (i.e., the DG molecules in the sphere body, plus DG molecules near or on the sphere surface).

Thanks for the valuable comments. I'm very sorry that we did not describe the experiment properly, which may have triggered some misunderstanding. The shortened lifetime data of DG molecules presented in the MS are only relevant DG molecules on the surface of the microsphere, but not for the entire DG molecules within the microsphere. The lifetime of DG molecules located at the center of the microsphere essentially remains unchanged.

We performed a more comprehensive measurement in order to answer the reviewer's question more clearly. The time-resolved lifetime measurement was performed using three parallel samples: DG microspheres in pure H₂O, DG microspheres in 100 μM R6G and DG microspheres in 100 μM R6GH. Regions of interest (ROIs) were selected as shown in figure 4 above. Lifetimes of DG molecules located at different regions of the microsphere (top, bottom, left, right, center and edge) were measured. For DG-H₂O and DG-R6GH samples, the fluorescence lifetimes of DG were measured consistently around 4.0-4.1 ns, regardless of the location of the ROI, suggesting that no energy transfer was taking place. However, for DG-R6G sample, shortened lifetimes were obtained for those DG molecules located in ROIs at the surface of the microsphere, suggesting FRET occurred; whereas for DG molecules located in the interior of the microsphere (center), the lifetime was not changed. These data suggest that FRET indeed occurred between surface DG molecules and R6G molecules.

We have modified our MS accordingly and included the data in revised MS, please refer to SI Figure S5.

Figure 4. Time-resolved fluorescence decay of DG molecules in the presence of R6G. **a.** Regions of interest (ROI) selection. The black-white striped areas indicate the cross-section of a DG doped microsphere. DG doped microspheres were ingrained in H₂O, R6G and R6GH respectively. The fluorescence decay was recorded using TCSPC as described in the experimental section via acquisition of 300000 photons. **b.** Fluorescence lifetime of each ROI (top, bottom, left, right, center and edge) under different conditions. The data in the table of each ROI is the mean value of five measurements. Since the FRET efficiency is directly dependent on the donor-acceptor distance and the orientation of the donor and acceptor molecules, the different lifetimes obtained from the different measurements indicate the diverse orientation R6G molecules and distances between DG and R6G molecules. **c,d & e.** Fluorescence decay curves of DG microspheres in H₂O, R6G and R6GH.

Minor

Figure S3. I am surprised that no lasing was observed when 15 μ m sodalime glass microsphere was embedded in the 50 μ M R6G solution. R6G lasing has been observed many times with ring resonators (microsphere based, capillary based, etc.).

Thanks for the comments. We agree with the reviewer. It is true that lasing can occur with ring resonators (capillary filled with dye) or microbeads embedded in dye solutions^{1,2}. In our MS, the no-lasing result obtained from the control experiment just demonstrated that with the existing setup, which was same as R6G lasing with FRET, we did not observe direct lasing from R6G with sodalime microsphere cavity. There might be a few reasons to explain the results:

First, the cavity size in our setup is relatively small (15 μ m) compared with the ring resonator in ref1 and ref2 below, in which the ring diameter was 75 μ m. As a result, higher pumping power might be required to get laser emission. Please note that, in figure S3, the pumping energy was

1262nJ; while in ref1 and 2, it was in μJ level. Indeed, when we increase R6G concentration to the mM-level and the pumping energy to the mJ-level, we observed R6G lasing pumped with 532nm nano-pulsed laser. Please see figure 5 below. Considering our setup aims to investigate intracellular applications, we prefer lower pumping energy to avoid cell damage.

Second, comparing with a similar setup in ref3, i.e. a microsphere embedded in a dye solution, indeed, they got lasing with a smaller cavity ($8.7\mu\text{m}$) and nJ pumping energy (ref3, figure 3e). We would say that the refractive index of the microbead in their setup is much higher ($n=1.96$) than our sodalime microsphere ($n=1.59$), which also favors lasing.

Under comparable conditions with our setup we do not achieve lasing in a sodalime microsphere in R6G, however it is possible under the same conditions to obtain lasing when the sodalime microsphere is replaced by a DG microsphere. This should in no way mean that there are no conditions under which the sodalime microsphere in R6G will result in lasing, it should just indicate that with the same conditions the lasing in the DG-R6G DOM system occurs more readily, indicating that much lower concentrations of R6G and lower pump lasing powers already yield R6G lasing.

References:

Ref1: Yuze Sun, Siyka I. Shopova, Chung-Shieh Wu, Stephen Arnold, and Xudong Fan, Bioinspired optofluidic lasers via DNA scaffolds, PNAS 2010 vol. 107 no. 37, 16039–16042

Ref2: Yuze Sun and Xudong Fan, Distinguishing DNA by Analog-to-Digital-like Conversion by Using Optofluidic Lasers, Angew. Chem. 2012, 124, 1262 –1265

Ref3: Matjaž Humar and Seok Hyun Yun, Intracellular microlasers, NATURE PHOTONICS | VOL 9 | SEPTEMBER 2015 | 572-576

Figure 5. Left panel: R6G lasing emission with a microsphere resonator. Sodalime microsphere was embedded in 20mM R6G pumped with 532nm pulsed laser of 1.26mJ. Right panel: Concentration-dependent wavelength shift.

2. Application

The paper shows the FRET lasing when a microsphere is internalized by a cell filled with R6G. This

application is less interesting and less practical. The best way is to demonstrate an actual sensing such as DNA hybridization on the microsphere surface or protein binding on the sphere surface.

We appreciate the reviewer's comments. It is true that real-time applications within a cell to reveal DNA hybridization or protein binding would be more exciting. At the moment, we are working toward this end and looking forward to the results (we are trying to sense intracellular microRNA molecules using the concept of FRET-WGM). Since the workload is heavy, including modifying the cavity surface with a recognizing molecule for specific binding, optimizing the concentration of the recognizing molecule, optimizing the optical conditions etc., in this MS, we only aimed to focus on other aspects than the intracellular applications, such as investigating the concept of FRET-WGM, the sensing mechanism, factors influencing the sensitivity and so forth. At end of the MS, we showed a brief intracellular measurement of R6G, only to demonstrate the feasibility of intracellular applications of the concept. We hope to show our intracellular sensing in the near future.

Finally, we would like to thank the referee again for taking time to review our manuscript and raising valuable concerns.

Reviewer #3 (Remarks to the Author):

This is a very interesting manuscript that reports on using wgm FRET lasing with DG donor dye molecules doped into a polystyrene wgm microspheres and R6G acceptor molecules at or near the wgm microsphere in solution, for detecting rhodamine 6G at μM concentration levels, in situ and inside of single cells. The detection is achieved by monitoring the wavelength blue and red shifts of the DG and R6G laser lines, respectively. Convincing evidence is provided for the FRET mechanism generating the lasing spectra and the wavelength shifts which are used for the sensitive (μM) detection of R6G with extrapolated sensitivity approx. few nM R6G, which is excellent for this approach.

Please consider revising the manuscript according to the following comments:

In their introduction, the authors could also mention other techniques that already detect /image even single fluorescent dye molecules inside of single cells, and how one compares these existing techniques to this novel work on intracellular wgm sensing of R6G.

Thanks for the constructive comment. Following the reviewer's advice, we have modified the introduction section of our MS via adding/comparing with the existing techniques. Please refer to the first paragraph of introduction in the revised MS.

In the discussion, the authors could mention how detection on their platform can be made specific to biomolecules of interest such as proteins, dna etc.

Thanks for the constructive comments. We have added detailed description of how specific detection of miRNA or protein molecules can be realized using the FRET-WGM platform, please refer to the legend of figure 6 in the revised MS.

"near-universal feature of these precision optical biosensors is that high light intensities" : I believe that this statement is incorrect for wgm sensors. The coupling of a high laser light intensity into (passive) wgm sensors does not directly result in any improvements of the molecule detection sensitivity. The wgm interferometric signal of a passive wgm sensor does not directly increase with increasing the laser power coupled to the wgm. Hence, the authors may wish to rephrase their statement that the precise molecule sensing with wgm depends on high light intensities. The reader should not confuse the local enhancements of near fields, which have been used in wgm sensing, with an incorrect claim that an overall high light intensity is required for the ultra-sensitive wgm sensing of molecules.

Thanks to the reviewer for pointing out the inappropriate statement in our MS. We are very sorry for the inaccuracy. We have deleted the incorrect statement and rephrased the text in the revised MS. Please refer to page 4, the first paragraph.

I suggest that the authors include (more) references for the observation of photo damage in passive wgm biomolecular sensing.

Thanks for the advice. We have rephrased our MS via deleting the incorrect statement about photo damage in WGM biomolecular sensing. So, references may not be necessary to include in the MS. For general interest, we find some references on photo damage in biosensing:

Wäldchen, S., Lehmann, J., Klein, T., Van de Linde, S. & Sauer, M. Light-induced cell damage in live-cell super-resolution microscopy. *Sci. Rep.* 5, 15348 (2015).

Landry, M. P., McCall, P. M., Qi, Z. & Chemla, Y. R. Characterization of photoactivated singlet oxygen damage in single-molecule optical trap experiments. *Biophys. J.* 97, 2128–2136 (2009).

Mirsaidov, U. , Timp, W. , Timp, K. , Mir, M. , & Timp, G. . (2008). Optimal optical trap for bacterial viability. *Physical Review E Statistical Nonlinear & Soft Matter Physics*, 78(2 Pt 1), 021910.

Sowa, Y. , Rowe, A. , Leake, M. , Yakushi, T. , Homma, M. , & Ishijima, A. , et al. (2005). Direct observation of steps in rotation of the bacterial flagellar motor. *Nature*, 437(7060), 916.

The authors may also like to compare the peak power densities encountered in the wgm evanescent field used for (fluorescent-free) molecule sensing to the peak power densities encountered in the focus of their pump beam that excites the active wgm microsphere cavities. The authors could then also compare all of these power density levels to the levels that are known to cause photo damage to dye molecules in (single-molecule) microscopy. I also note that their technique does rely on excitation of dye molecules which are naturally more prone to photo damage as compared to non-fluorescent molecules that are detected on a label free, passive wgm sensor.

Thanks for the comments. We appreciate the reviewer's comprehensive advice, which would be very helpful in investigating the effects of photo damage. This will be very interesting to pursue in the near future. In this MS, our focus is intracellular application. Throughout the experiments, we did not observe cell death after numbers of measurements, indicating the pumping energy is not harmful to cells. Since the light is focused on the center of the microsphere, we do observe destruction of the microsphere under extreme pumping conditions, such as extremely high pumping energy (1.5 μ J). However, most of the time, we keep the pumping energy at moderate level.

The proposed technique described here does rely on excitation of dye molecules that are naturally more prone to photo damage, which might be a downside of active mode WGMs. In our experiments, we do observe photo bleaching of the dye when the microsphere is continuously pumped for 10min, though, the data acquisition process only takes a couple of minutes, far before severe photo bleaching occurs. If photo bleaching is a problem hindering the experiment, there are ways to solve the problem. For example, the microcavity can be fabricated using more stable fluorescent materials such as quantum dots, light-emitting semi-conductor materials, or perovskite etc.

Generally speaking, a passive-mode WGM is a good technique for ultra sensitive sensing for the level of down to single molecule detection, especially with the aid of sensitivity-enhancing techniques. The unique advantages of a passive WGM is beyond comparison with an active wgm. The major limitation that obstructs its use in intracellular applications is that it requires an additional coupler. If this can be overcome in future, a passive WGM would be more prospective for biological applications.

In respect to active WGMs, due to its limited spectral resolution, the sensitivity is not as good as that of a passive WGM. Also, because of using laser light to excite the dye, the light intensity

needs to be carefully optimized to avoid photodamage to the cell as well as photo bleaching of the dye. The major advantages of the technique we describe in our MS are (1) the described setup allows for free space excitation and (2) the sensing performance is greatly enhanced due to the integration of FRET, enabling intracellular applications. In addition, the narrower laser linewidth compared to conventional fluorescence emission allows for a more accurate quantification of the concentration levels of the analyte. Furthermore, the proposed technique uses the wavelength as a measure to quantify the concentration of an analyte. In this case, even if, to some extent, photo bleaching occurs, the wavelength remains stable (as demonstrated in the MS), which ensures the precision of quantification even in this cases. Of course, every technique has its benefits and downsides. The technique we described here is no exception, for example, suitable FRET pairs need to be carefully selected, photo bleaching may occur and the detection limit needs to be further decreased, and so forth.

Overall, I do not understand why the authors motivate their nice work with a problem of photodamage in wgm sensing. In my opinion, the main motivation for biosensing with active cavities is the in vivo sensing applications that are made possible by the free space excitation with the pump light. In my opinion, photo damage can only be made worse with the use of active cavities because of the requirement of pulsed laser excitation, dye molecules, and comparatively high peak power densities.

Thanks to the reviewer for the constructive comments. We agree with the reviewer that one big advantage of active microcavities is the employment of free space excitation that makes it more suitable for in-vivo sensing. Although we do not observe any photo damage to the cell since the pumping power is very low (nJ-level, the cell remains alive for at least 72 hours following excitation), we realize that photo damage indeed is not a very good motivation for the work. Following the reviewer's advice, we have rephrased our motivation in the introduction section, please refer to the introduction of revised MS.

"As the polystyrene microsphere is negatively charged": can the authors elaborate and provide experimental evidence for this negative charge? What is the buffer pH, pK of the chemical groups that carry the negative charge, etc.?

Thanks for the constructive comments. Following the reviewer's advice, we have included experimental evidence showing the charging properties of the microspheres and molecules. The measured zeta potential of a DG doped microsphere is -11.3mV, which was determined using a Zetasizer Nano ZS. For R6G molecules, the zeta potential is +14.2mV, please see the figure below. The positive charging of R6G is also well documented in publications [ref1].

We also determined the zeta potentials of sodalime microsphere and R6GH molecules to be -9.85mV for sodalime microspheres and +16.3mV for R6GH molecules, respectively. The data indicate that sodalime microspheres are negatively charged, similar to DG-doped PS microspheres; whereas R6GH molecules are positively charged, similar to R6G molecules.

The microspheres and R6G were suspended or dissolved in H₂O to keep consistence with our data

described in the MS. R6GH was dissolved in 90%H₂O+10% EtOH. We have included the data in our revised MS, please refer to SI figure S4.

DG microsphere

Results			
	Mean (mV)	Area (%)	St Dev (mV)
Zeta Potential (mV): -11.3	Peak 1: -11.3	100.0	4.60
Zeta Deviation (mV): 4.60	Peak 2: 0.00	0.0	0.00
Conductivity (mS/cm): 0.00589	Peak 3: 0.00	0.0	0.00
Result quality Good			

sodalime microsphere

Results			
	Mean (mV)	Area (%)	St Dev (mV)
Zeta Potential (mV): -9.85	Peak 1: -9.85	100.0	3.55
Zeta Deviation (mV): 3.55	Peak 2: 0.00	0.0	0.00
Conductivity (mS/cm): 0.0151	Peak 3: 0.00	0.0	0.00
Result quality See result quality report			

Results			
	Mean (mV)	Area (%)	St Dev (mV)
Zeta Potential (mV): 14.2	Peak 1: 14.2	100.0	7.11
Zeta Deviation (mV): 7.11	Peak 2: 0.00	0.0	0.00
Conductivity (mS/cm): 0.00334	Peak 3: 0.00	0.0	0.00
Result quality Good			

Results			
	Mean (mV)	Area (%)	St Dev (mV)
Zeta Potential (mV): 16.3	Peak 1: 16.3	100.0	17.7
Zeta Deviation (mV): 17.7	Peak 2: 0.00	0.0	0.00
Conductivity (mS/cm): 0.00668	Peak 3: 0.00	0.0	0.00
Result quality Good			

Figure 1. Charging properties of the microspheres, R6G and R6GH molecules. Data were measured on a Zetasizer Nano ZS (Malvern Panalytical)

References:

[1] Gear, A. R. L. . (1974). Rhodamine 6g: a potent inhibitor of mitochondrial oxidative phosphorylation - sciencedirect. Journal of Biological Chemistry, 249(11), 3628-3637.

“Here, we demonstrate that incorporating Förster resonance energy transfer (FRET) to a WGM microcavity platform can greatly increase the sensitivity” I would ask the authors to be more specific about which version of the wgm sensor they talk about. This statement is generally speaking incorrect, I believe.

Thanks to the reviewer for the constructive comments. We have rephrased the sentence as: “Here, we demonstrate that incorporating Förster resonance energy transfer (FRET) to an active-mode WGM microcavity platform can greatly increase the sensitivity”. We also checked the whole MS to rephrase the text accordingly.

It is indeed very interesting in wgm FRET lasing sensing that increasing the concentration of acceptor molecules at uM concentration levels (where I assume any adsorption of the molecule to the sensor is always at saturation) results in blue shift of the donor lasing lines and a red shift for acceptor lasing lines. This makes for a very interesting and novel detection mechanism. In their control experiments, how do the authors know that the non fluorescent R6GH molecule

adsorbs in the same way to the microsphere surface as R6G?

Thanks for the constructive comment. To address the reviewer's concern, we performed zeta potential experiments to confirm this. The data suggest that both DG doped PS microspheres and non-fluorescent sodalime microspheres are negatively charged. The zeta potentials for DG microspheres and sodalime microspheres are -11.3mV and -9.85mV, respectively. The zeta potential of R6G and R6GH in ddH₂O are measured to be +14.2 mV and +16.3 mV, respectively, indicating both of them are positively charged, please refer to figure 1 above. So R6GH molecules adsorb on the DG doped PS microspheres and sodalime microspheres the same way as R6G. . Therefore, R6GH molecules interact with the DG microspheres and sodalime microspheres in the same way as R6G molecules.

Why have the authors not varied the microsphere surface charge to test their hypothesis that surface adsorption of R6G to the native charged microsphere is required for FRET lasing to occur?

Thanks for the constructive comment. Following the reviewer's advice, we have modified the microsphere surface charge to further confirm our hypothesis.

We altered the surface charge of the microsphere to abolish the distance requirement for FRET to occur. To this end, we used PAH molecules (Poly (allylamine hydrochloride), MW 10000-20000 Da) to coat the microspheres in order to achieve a positively charged surface. TEM image confirmed a ~5nm layer formed on the surface of the microsphere (figure 2a & b below) and the measured zeta potential of DG microsphere after PAH modification was +15.9mV (figure 2c). The positively charged microsphere was embedded in 50μM R6G solution and pumped with a 473nm pulsed laser. We randomly selected five microspheres and the data are shown in figure 2d. We observed solely DG lasing for the PAH coated DG microspheres instead of DG and R6G dual lasing that we observed for the uncoated DG microspheres. The CWL of the DG lasing is 535.8nm, which is comparable with the lasing wavelength we observed in pure H₂O (red-shifted by 1.8nm due to a locally increased refractive index owing to the PAH modification). It is assumed that the positively charged DG microspheres repel R6G molecules and thus R6G molecules have a larger distance from the PAH-coated microsphere, interfering with the energy transfer via FRET, which strongly depends on the distance between donor and acceptor molecules. Therefore. R6G fails to lase under this condition. The data further supports the mechanism of FRET playing the leading role in the FRET-WGM sensing system.

We have added the data to the revised MS, please refer to page 15 first paragraph and SI figure S9.

Figure 2. Altering the surface charge of the microsphere to abolish FRET. **a & b.** TEM images show the morphology of a microsphere prior (**a**) and post (**b**) PAH-coating. **c.** Zeta potential of PAH-modified microspheres is 15.9mV. **d.** Lasing spectra after surface charge alteration. A PAH-modified DG microsphere was embedded in 50 μ M R6G solution and pumped with 473nm laser. Five microspheres were randomly selected and the lasing spectra are displayed. Only DG lasing, with a CWL @535.8nm, rather than DG and R6G dual lasing were observed, showing that the FRET effect was abolished by the surface modification.

Have the authors considered that the slight heating of the microsphere may play a role in the shifts of the donor and acceptor lasing lines? How does the shift depend on the laser power/intensity?

Thanks for the advice. Following the suggestion, we performed an experiment under higher temperature. A DG-doped microsphere was embedded in 75 μ M R6G and excited with a 473nm laser. We found that when the temperature increased to 30 $^{\circ}$ C, the lasing wavelengths of DG and R6G both shifted toward the shorter wavelength region, compared with the room temperature results (20 $^{\circ}$ C), as shown in the following figure 3a (Please refer to SI fig.S11 for full lasing spectrum). The blue shift of DG and R6G resonant peaks can be explained by Wien's displacement law, *i.e.* the emission wavelength shifts towards the short wavelength region when the temperature increases. In addition, we observed that the wavelength gap between R6G and

DG ($\Delta\lambda=\lambda_{R6G}-\lambda_{DG}$) remained nearly same at the higher temperature, with $\Delta\lambda_{30}=45.17\text{nm}$, compared to $\Delta\lambda_{20}=45.52\text{nm}$ at room temperature. The lasing thresholds at higher temperatures for both DG and R6G are higher than at room temperature, with 309nJ vs. 209nJ for DG; and 339nJ vs. 309nJ for R6G, respectively (c.f figure 3b). The resonant wavelength at 30 °C also remains stable upon extended pumping time, figure 3c. The data indicate that the ambient temperature is not a key factor influencing the energy transfer via FRET.

To answer the second part of the reviewer’s question, under increased pumping laser power, the wavelength did not shift, instead, we observed only intensified emission peaks, as shown in figure 3d.

Figure 3. Effect of temperature on FRET-WGM sensing. **a.** CWL of DG and R6G lasing at temperature of 20°C and 30°C, respectively. The higher temperature resulted in a blue-shift of the resonance wavelength. Please refer to SI fig.S11 for full lasing spectrum. **b.** Lasing thresholds of DG and R6G at 30°C are 309nJ and 339nJ, respectively. **c.** At higher temperature, the resonance wavelengths of DG and R6G remain stable for increased pumping time. **d. Effect of pumping energy on resonant wavelength.** The resonant wavelength did not shift under increased pumping power, only peaks intensified.

We have included the data in the revised MS, please refer to page 16, first paragraph and SI figure S11.

The R6G redshift could be due to increase of refractive index in the higher R6G solution concentration? Can this be ruled out?

Thanks for the comments. In the non-FRET WGM system (pure WGM), the refractive index

change is the dominant factor for the observed wavelength shifts. We measured the refractive indexes of three concentrations of R6GH solutions. The data are shown in the table below. We can see that refractive index difference of these three solutions is very small; correspondingly, therefore the response of the microcavity ($\Delta\lambda$) is negligible.

While the increase of refractive index of R6G for the three measured concentrations is comparable to that of R6GH, as shown in the table, the observed wavelength shift for the resonant peaks of DG and R6G $\Delta\lambda(\lambda_{R6G}-\lambda_{DG})$ is quite considerable, indicating that refractive index change is not the determining factor in this case. In contrast, the resonant energy transfer plays a major role under these conditions. This conclusion is further supported by the energy transfer efficiency data. By measuring the fluorescence lifetime of DG in the presence or absence of R6G, we calculated the FRET efficiency E. The maximum E value is as high as 58%, which is much higher than most published data (ref1-4). In addition, when the microcavity is pumped, the evanescent field around the cavity enhances the fluorescent energy transfer [ref5-7], which might contribute to the enhanced sensitivity of FRET-WGM and also explains the extraordinary FRET efficiency.

For more comprehensive insights into the microcavity energy transfer, we applied rate equations to simulate the intensity for every single mode and thereby obtained the efficiency of the cavity energy transfer. Please refer to SI section 3.

Table: Refractive index of different concentrations of R6G and R6GH.

R6G concentration	refractive index (n)	Δn (n-n0)
0 (H2O)	1.332	
10nM	1.3351	0.0029
100nM	1.3354	0.0032
10uM	1.3361	0.0039
R6GH concentration		
0 (90%H2O+10%EtOH)	1.3346	
10nM	1.3367	0.0021
100nM	1.3375	0.0029
10uM	1.3382	0.0036

note: R6G and R6GH are dissolved in H2O and 90%H2O+10%EtOH, respectively.

References:

- [1]. Chakraborty U, Maiti P, Singha T, et al. Effect of montmorillonite clay on the fluorescence resonance energy transfer between two cationic dyes Acridine Orange and Rhodamine B in solution[J]. Materials today: proceedings, 2020.
- [2]. Chakraborty S, Arshad Hussain S. Fluorescence resonance energy transfer (FRET) between acriflavine and CdTe quantum dot[J]. Materials Today: Proceedings, 2020.
- [3]. Zambrana-Puyalto X, Ponzellini P, Nicolò Maccaferri, et al. Förster-Resonance Energy Transfer between Diffusing Molecules and a Functionalized Plasmonic Nanopore[J]. Physical Review Applied,

2020.

[4]. Pramanik A, Biswas S, Sekhar Tiwary C, et al. Forster resonance energy transfer assisted white light generation and luminescence tuning in a colloidal graphene quantum dot-dye system[J]. Journal of Colloid and Interface Science.

[5]. Andrew, P. Frster energy transfer in an optical microcavity.[J]. Science, 2000, 290(5492):785-788.

[6]. Xiaolan, et al. "Non-Radiative Energy Transfer Mediated by Hybrid Light-Matter States." Angewandte Chemie International Edition 55.21(2016):6202-6206.

[7]. Hertzog M, Mao W, Mony J, et al. Strong light-matter interactions: A new direction within chemistry[J]. Chemical Society Reviews, 2019, 48(3).

“increased losses lead to a frequency shift towards shorter wavelengths” ... perhaps the authors might want to elucidate this mechanism a bit more... I am not sure if the interdisciplinary readership of this journal can follow their argument.

Thanks for the comments. We have elaborated a bit more in the revised MS. We also cited one relevant reference for the readers who are interested in the fundamentals.

Can the authors comments on the fundamental problem of using active wgm cavities for ultra-sensitive detection of gain molecules as this will always required a minimal concentration of gain molecules? Can one estimate this minimal concentration of dye molecules that is required to build a FRET laser which I assume also depends on the Q factor?

Thanks for the comments. When using active-mode WGM microcavity for sensing, there are several issues to consider:

1. Photo bleaching which we have discussed above.
2. Initial concentration of the donor. The initial concentration of DG will determine the absolute value of CWL and sensitivity to R6G concentration. We have performed additional experiments and the results were discussed in the revised MS, please refer to page 18-19. Basically, increasing the initial concentration of the donor leads to a lowered limit of detection (LOD).
3. Cavity size. A larger cavity is less sensitive due to the surface density change of the acceptor. In addition, because most mammalian cells are between 10 to 100 μm in diameter, a smaller cavity would be preferential for intracellular applications. However, reducing the radius of the microsphere ultimately leads to a decrease in the Q-factor owing to diffraction as well as other size-dependent losses. So the optimization of the cavity size is a trade-off between sensitivity and practicality.
4. Q-factor. The lowest acceptor density that can be detected for a given active-mode WGM system depends on the resonance line width $\delta\lambda$, which is determined by the Q-factor; $\delta\lambda = \lambda / Q$. The larger Q, the smaller the dissipation and the lower the surface density that can be detected.
5. Quantum yield (QY) of the gain molecule. A high QY of donor and acceptor would favor lasing.

The minimal concentration of gain molecules required for sensing can be optimized experimentally. Reducing the number of donor molecules results in a higher lasing threshold for both donor and acceptor. Fewer donor molecules mean less energy is transferred to the acceptor molecules; therefore, the lasing threshold for the acceptor would also be increased. It is necessary to optimize

the ratio of donor and acceptor molecules to realize dual lasing as well as to achieve sensitive detection.

Finally, we are grateful to the reviewer for taking time to review our manuscript and raising valuable concerns. Thanks again.

Reviewers' comments:

Reviewer #1 (Remarks to the Author):

The revised manuscript provides additional data to support that the lasing mechanism is based on FRET. However, it fails to address two major points.

First, it is still lacking in a clear (quantitative) explanation for the high sensitivity to the concentration of the acceptor dye. The rate equations are described without their solutions or numerical simulations. The model does not describe the spectral states of the donor and acceptor molecules. Therefore, I doubt that the simple rate equations could explain the spectral changes. It may not be necessary to have a full theory or simulation, but some quantitative estimates that agree with the experimental result will be helpful as a scientific validation as well as a guide for the reader to apply the principle for different dyes.

Second, despite some new graphs included in the revision I remain unconvinced that the CWL or "mean wavelength" yield reliable measurement at low concentration. Why should the CWL be a continuous function of concentration? I expect the maximum intensity peak to hop from one mode to another as the concentration changes and as the spectral envelope shifts. The peak may appear to move continuously when the increment of concentration is relatively large so that the wavelength change is larger than the mode spacing.

I am particularly concerned about Fig. 3a in comparison to the added unprocessed data in Fig. S7. First, the authors did not include unprocessed spectra between 0 and 1 μM . Seeing the highly multi-mode spectra spanning over 20 nm, it is not clear at all how the authors were able to separate the lasing spectra of the two dyes (if they are separate) and determine the CWL of each for the low concentrations less than 1 μM at which $\Delta\lambda$ is claimed to be less than 20 nm.

Without adequate feasibility data, I think Fig. 6 should be removed or moved to the supplemental information.

Reviewer #2 (Remarks to the Author):

The authors reported FRET-based whispering gallery mode (WGM) lasing as a sensing platform, particularly for intracellular sensing.

The microsphere is first doped with donor dyes (such as Dragon Green) and then it is immersed in the acceptor solution such as R6G solution. FRET occurs between the donor dye and the acceptor dye near the microsphere surface. Upon excitation of the donor dye, donor dye lases. When there are acceptor dye molecules near the sphere surface, the FRET is able to pump the acceptor dye for it to lase at a longer wavelength. In the meantime, the donor dye lasing threshold increases (due to the loss to the acceptor dye). With the increased concentration of the acceptor dye, the donor lasing wavelength shifts to a lower wavelength (due to more loss to the acceptor) and meanwhile the acceptor dye lasing wavelengths shifts to a higher wavelength (due to more gain in the acceptor laser system since the acceptor concentration is higher). The spectral gap between the donor lasing wavelength and the acceptor lasing wavelength is then used as the sensing signal. For example, at 1 μM of R6G, the spectral gap between the Dragon Green (donor) lasing wavelength and the R6G (acceptor) lasing wavelength can be 12-20 nm. The authors use the lasing linewidth as the spectral resolution and then estimate the detection limit is about 60 pM of R6G. Finally, the authors use this FRET-WGM sensor to detect the R6G concentration inside cells and 3 μM R6G concentration is experimentally detected.

While the FRET-WGM lasers have been demonstrated many times in the past ten years (see the Refs. 2-5 given below and the authors should cite some of those papers), it is nice to see the authors to use the spectral gap as the sensing mechanism. The authors did thorough experiments to prove

the FRET is the mechanism behind the acceptor lasing and is responsible for the spectral gap.

However, there are a few major flaws and drawbacks in this work.

1. The authors introduced a dual core-shell structured ring resonators, one is formed by the microsphere itself and the other one is formed by a layer for R6G molecules near the surface. The evidence is that (1) the free-spectral-range analysis shows the ring diameter for R6G is slightly larger than the ring diameter for Dragon Green. (2) The image. However, the small refractive index contrast between the layer of R6G and the surrounding solution (water + a certain amount of R6G in free solution) may not be able to support another ring resonator. What really matters in the free spectral range calculation is $n \times D$ (refractive index times diameter). The decreased FSR can be contributed to the increased refractive index near the sphere surface. The Dragon Green sees a little bit refractive index increase due to the layer of R6G. Most of the Dragon Green is still inside the microsphere. For R6G, its WGM is pulled outward slightly more than Dragon Green (since the gain position is a little different. The gain for R6G is a little bit more outside than Dragon Green). In any case, whether there is a dual-resonator does not affect the sensing mechanism and the sensor performance.

2. On page 10 (Eq. 1 and the paragraph below). Eq. 1 explains so-called reactive sensing mechanism (or passive sensing). It is applicable to non-FRET-WGM, but cannot be applied to the active sensing. The explanation in the paragraph below Eq. 1 is incorrect (later, the authors use the coupled rate equations to explain the spectral gap between the donor lasing wavelength and the acceptor lasing wavelength – that is correct). Besides, the authors do not give detailed description of $E_x(r_0)$. The plasmonic effect the authors cited in the same paragraph is reactive sensing, to which Eq. 1 is applicable.

3. The mechanism behind the spectral gap between the donor lasing and acceptor lasing is well known. An earlier paper was published in 2000 (see Ref. 1 below). Later, there are a few more papers published to explain in details the FRET laser and the lasing wavelength in the FRET laser (see Refs. 2-5). While the rate equation is nice, but simple absorption and emission cross section analysis may be more intuitive. In short, when the acceptor concentration increases, the donor loss increases. The absorption/emission cross section analysis shows the shorter wavelength is easier to lase. That's why the donor lasing wavelength shifts towards the shorter wavelength. Similarly, when the acceptor concentration increases, the acceptor gain becomes higher. The same absorption/emission cross section analysis shows the acceptor lasing becomes easier at a longer wavelength. That's why the acceptor lasing wavelength shifts to the longer wavelength. Therefore, a few more citations should be included to cover the FRET lasing and the lasing wavelength shift mechanism (for example, Refs. 1-5).

4. The authors claim that 60 pM R6G detection limit is possible. This is misleading. First, this FRET-WGM sensing system is not linear and the slope is very shallow. For example, at 1 μ M of R6G (Table 1), the spectral gap is 12.8 nm. At 0.01 μ M, the spectral gap is 6.15 nm. The concentration drops by 100 fold, but the spectral gap decreases only 50%. Second, there is a concentration floor for R6G below which no lasing can be achieved. Without lasing, there is no spectral gap. The lower floor of R6G is on the order of 1-10 μ M. Third, there is a floor for the spectral gap. For example, the lowest lasing wavelength may be, say, 532 nm. You cannot push the spectral gap below a certain number, for example 1 nm. I guess

For clarity, it is suggested the authors to plot Figure 3b and c in the log-log scale.

5. Relying on the electrostatic interaction to pull more R6G towards the sphere surface is a little risky, as it affects the quantitation of the R6G concentration (or any other analyte concentration). In reality, if the ionic strength in solution changes, the R6G surface density may change quite a bit.

1. Moon et al., Phys. Rev. Lett. Vol. 85, 3161 (2000).
2. Sun et al., PNAS Vol. 107, 16039 (2010).

3. Sun et al., Angew. Chem. Intl. Ed. Vol. 51, 1236 (2012)
4. Chen et al., Lab Chip Vol. 13, 3351 (2013)
5. Chen et al., Lab Chip. Vol 16, 2228 (2016)

Reviewer #3 (Remarks to the Author):

The authors have addressed my previous comments.

The discussion of the sensing mechanism which explains the observed blue shift of the resonance wavelength should be handled carefully, especially because the exact cause of these wavelength blue shifts may not yet be entirely understood. Please carefully check all statements in that regard such as:

“The energy loss of GG molecules marks a blue shift of the resonance wavelength”

It would seem to me at first that optical loss leads to a change in resonance linewidth and lasing threshold, not frequency.

Brief response letter

Replies to Reviewer 1

Reviewer #1 (Remarks to the Author):

The revised manuscript provides additional data to support that the lasing mechanism is based on FRET. However, it fails to address two major points.

First, it is still lacking in a clear (quantitative) explanation for the high sensitivity to the concentration of the acceptor dye. The rate equations are described without their solutions or numerical simulations. The model does not describe the spectral states of the donor and acceptor molecules. Therefore, I doubt that the simple rate equations could explain the spectral changes. It may not be necessary to have a full theory or simulation, but some quantitative estimates that agree with the experimental result will be helpful as a scientific validation as well as a guide for the reader to apply the principle for different dyes.

Response: We appreciate the reviewer's insightful suggestion. As far as the theoretical scheme is concerned, although not presented in this MS, we have been working towards it. We think a theoretical model under the steady-state condition can be developed based on the rate equations and absorption/emission cross section analysis of the acceptor and donor. Here we would like to describe what we proposed to do briefly as follow (we have included the following content in the revised MS, please refer to section 4 of SI):

As we described in the previously revised MS, the acceptor-based laser rate equation is shown in Eqn. (1):

$$\frac{dq_a(t)}{dt} = \frac{cq_a(t)n_a}{n_2} \sigma_{ea}(\lambda) - \frac{cq_a(t)(N_a - n_a)}{n_2} \sigma_{aa}(\lambda) - \frac{q_a(t)}{\tau_{ea}} \quad (1)$$

Where N_a is the total concentration of acceptor molecules, and n_a is molecular density of acceptor dyes in the excited state. $\sigma_{ea}(\lambda)$ and $\sigma_{aa}(\lambda)$, respectively, are the acceptor emission and absorption cross-section. q_a is the acceptor emitted photon density. τ_{ea} is the lifetime of the photons in the cavity, which equals to $\eta Q \lambda / 2\pi c$. n_2 and c represent the refractive index of the surrounding medium (~ 1.335) and light speed in the vacuum, respectively. η means the fraction of mode energy in the evanescent field. Under steady-state conditions, Eqn. (1) can be written to Eqn. (2):

$$n_a \sigma_{ea}(\lambda) - (N_a - n_a) \sigma_{aa}(\lambda) - \frac{2\pi n_2}{\eta Q \lambda} = 0$$

(2)

Therefore, from Eqn. (2), we can obtain the fraction of acceptor molecules at the excited state under the threshold condition [ref1]:

$$\gamma_{tha} = \frac{n_a}{N_a} = \frac{1}{\sigma_{ea}(\lambda) + \sigma_{aa}(\lambda)} \left[\sigma_{aa}(\lambda) + \frac{2\pi n_2}{\eta Q \lambda N_a} \right] \quad (3)$$

Where γ_{tha} represents the lasing threshold of acceptor dye.

Similarly, the donor-based laser rate equation can be described in Eqn. (4).

$$\frac{dq_d(t)}{dt} = \frac{cq_d(t)n_d}{n_1} \sigma_{ed}(\lambda) - \frac{cq_d(t)(N_d - n_d)}{n_1} \sigma_{ad}(\lambda) - \frac{q_d(t)}{\tau_{ed}} + \frac{cq_d(t)n_a}{n_2} \sigma_{ea}(\lambda) - \frac{cq_d(t)(N_a - n_a)}{n_1} \sigma_{aa}(\lambda) \quad (4)$$

Where N_d is the total concentration of donor molecules, and n_d is molecular density of donor dyes in the excited state. $\sigma_{ed}(\lambda)$ and $\sigma_{ad}(\lambda)$, respectively, are the donor emission and absorption cross-section. q_d is the donor emitted photon density. τ_{ed} is the lifetime of the photons in the cavity, which equals to $\eta Q \lambda / 2\pi c$. n_1 and c represent the refractive index of the microcavity (~ 1.59) and light speed in the vacuum, respectively. η means the fraction of mode energy in the evanescent field. Under steady-state conditions, Eqn. (4) can be written to Eqn. (5):

$$n_d \sigma_{ed}(\lambda) - (N_d - n_d) \sigma_{ad}(\lambda) - \frac{2\pi n_1}{\eta Q \lambda} + \frac{n_1 n_a}{n_2} \sigma_{ea}(\lambda) - (N_a - n_a) \sigma_{aa}(\lambda) = 0 \quad (5)$$

From Eqn. (5), we can obtain the fraction of donor molecules at the excited state under the threshold condition:

$$\gamma_{thd} = \frac{n_d}{N_d} = \frac{1}{\sigma_{ed}(\lambda) + \sigma_{ad}(\lambda)} \left[\sigma_{ad}(\lambda) + \frac{2\pi n_1}{\eta Q \lambda N_d} + \frac{(1 - \gamma_{tha}) \sigma_{aa}(\lambda) N_a}{N_d} - \frac{\gamma_{tha} N_a \sigma_{ea}(\lambda)}{n_2 / n_1} \right] \quad (6)$$

Where γ_{thd} represents the lasing threshold of the donor dye.

Please note that, as far as we know, the donor-based equations have never been described in previous publications.

According to Eqn. (3) and Eqn. (6), different R6G acceptor concentrations will give to corresponding γ_{th} values of donor or acceptor at respective wavelengths. So when pump energy reaches the threshold (γ_{th}), donor or acceptor will lase at a particular wavelength. Since the absorption and emission cross sections are determined by the dye concentration as well as the energy gain or loss via FRET, also, the emission and absorption cross sections are a continuous function of wavelength, thus, the thresholds will be a function of the wavelength (other parameters can be regarded as fixed values). Therefore, under different acceptor concentrations that energy transfer is different, the laser threshold and resonance wavelength will be different, i.e. the threshold and resonance wavelength are dependent on the acceptor concentration. Therefore, theoretically, we could simulate and calculate the concentration-dependent shifts of the resonance wavelengths of both donor and acceptor. The relationship between concentration and wavelength gap would then be constructed.

Once the theoretical model is done, it will be of great significance in predicting the donor-acceptor wavelength gap at a certain acceptor concentration and thus theoretically explain the sensing mechanism. I would say that the proposed theoretical work cannot be accomplished overnight. It requires the measurement of absorption and emission cross sections at various concentrations, determining the energy transferring efficiency at each concentration and loads of calculations. Hopefully, we can work out ASAP and publish the data in future.

Nevertheless, we would like to say that the focus of this work was on the demonstration of the feasibility to intracellular application. Recent years, life scientists have come to realize that

individual cells can differ dramatically in size, protein levels, and expressed RNA transcripts. These variations are key to answering previously unsolvable questions in the life science area. Single cell analysis can avoid the mistake of taking averages of entire cell populations, discover previously undetected subpopulations and unveil new regulatory path. However, due to the extremely low molecular content within a single cell, intracellular molecular studies at single cell resolution is extremely difficult. Given the importance of single cell analysis, our proposed approach would be particularly attractive to the life scientists in the current single cell era. In this context, our proposed technique that provides a new approach for single cell analysis is of great value. We thought this was a kind of breakthrough in the area of intracellular molecular analysis at single cell resolution. As far as we know, this will be the first report of WGM sensing for intracellular analysis at molecular level. Previously, there have been several exciting reports (*Nat Photonics*. 2015 9(9):572; *Nat Photonics*. 2019 13(10):720; *Light Sci Appl*. 2021 25;10(1):23; *Nat Commun*. 2018 16;9(1):4817; *Nat Photonics*. 2020 14 (7) ,452) showing intracellular WGM applications. Their work was mainly focused on cellular behavior, such as tracking cell movement via WGM labeling, monitoring subtle changes of intracellular microenvironment etc., *i.e.*, more focused on information acquisition at cellular level. In terms of this, we would say that construction of a theoretical model might be beyond the scope of the MS. The current data are original, sufficient and complete.

With regard to providing guidelines for the follow-on readers to choose different dyes, I would say that the technique can be applied with any commonly used FRET-pairs as long as they meet the FRET requirements, for example, fluorescent dye pairs Cy3-Cy5 or FAM/TAMRA, and fluorescent protein pairs GFP-YFP that can be co-expressed with proteins of interest. A big advantage of the proposed technique is easy to follow. It is not like plasmonic technique that requires precise coupling and extreme stability of the optical system.

In short, our findings revealed a new approach that can be used to increase the sensitivity of WGM and suitable for molecular analysis at single cell resolution.

Ref1. Chen, Y.-C.; Chen, Q.; Fan, X., Optofluidic chlorophyll lasers. *Lab Chip* 2016, 16 (12), 2228-2235.

Second, despite some new graphs included in the revision I remain unconvinced that the CWL or "mean wavelength" yield reliable measurement at low concentration. Why should the CWL be a continuous function of concentration? I expect the maximum intensity peak to hop from one mode to another as the concentration changes and as the spectral envelop shifts. The peak may appear to move continuously when the increment of concentration is relatively large so that the wavelength change is larger than the mode spacing.

Response: We appreciate the reviewer's insightful comments. We would say that, for the maximum intensity peak, it is mode shift rather than mode hop. We apologize for not describing clearly in the last round response letter. Here we would like to explain in more detail.

As we know, the theoretical resonant wavelength of DG-doped microcavity can be calculated

by using the following formula:

$$\lambda \cong \pi n_{\text{most}} D \left[v + \frac{\alpha_s v^{\frac{1}{3}}}{2^{\frac{1}{2}}} - \frac{P}{(n_{\text{eff}}^2 - 1)^{\frac{1}{2}}} + \frac{3}{10} \frac{\alpha_s^2}{2^{\frac{2}{3}} v^{\frac{1}{3}}} - \frac{P \left(n_{\text{eff}}^2 - \frac{2P}{3} \right)}{(n_{\text{eff}}^2 - 1)^{\frac{3}{2}}} \frac{\alpha_s}{2^{\frac{1}{3}} v^{\frac{2}{3}}} \right]^{-1}$$

Where λ is the resonant wavelength, R is the radius of the microsphere and m is mode number. $v = m + 1/2$, $n = n_1/n_2$, where n_1 and n_2 are the refractive indices of the microspheres and the environment solution. P is the polarization characteristic coefficient, $P = n$ for TE modes, and $P = n^{-1}$ for TM modes. $\alpha_l(\lambda)$ is the l th of the Airy function, where l is the radial mode number.

The refractive index of the microsphere is 1.59 (n_1). We first calculated the number of modes in correspondence to different lasing wavelengths of DG when the microcavity was embedded in pure water (refractive index is 1.33 (n_2)), as shown below in figure 1. The experimental and calculated values are in good agreement as shown in Table 1. It is found that when the DG microcavity was pumped in pure water, the TE mode is dominant.

Figure 1. Mode analysis of the DG-doped microcavity.

Table 1. Experimental and theoretical values of different modes.

Modes	135	134	133	132	131
Theoretical Value (nm)	531.8534	535.6812	539.5649	543.5058	547.5052
Actual Value (nm)	531.85005	535.73567	539.6628	543.58395	547.4991
$\Delta\lambda$ (nm)	0.00335	-0.05447	-0.097	-0.07815	0.00615

The wavelength of the maximum TE mode will be used as a basis for our judgment. The measured adjacent mode spacing (free spectral range) is shown in table 2. The FSR of DG is around 3.91nm in pure water. The FSR value is slightly different from one microcavity to another due to slight diameter variation caused by fabrication.

Table 2. Measured FSRs in between modes.

Modes	135-134	134-133	133-132	132-131
FSR (nm)	3.88562	3.92713	3.92115	3.9151

Keeping this in mind, we then can do spectral analysis of our data.

Here, we use the unprocessed data of low concentration range 0-1uM, also the major concern of the reviewer, for spectral analysis. First of all, we would like to clarify that, throughout our experiments, the maximum peak (CWL) consistently remains the maximum height in the spectral envelope under optimized excitation conditions, i.e., the pump energy is above the threshold but not far beyond the lasing threshold. So it can be easily identified without ambiguousness.

Figure 2 shows the original spectra of DG microcavity embedded 0, 10nM R6G, 50nM R6G, 100nM R6G, and 500nM R6G, respectively. At each concentration, we tested ten microcavities, and took the average as the final wavelength value. The maximum DG wavelength is @ 539.403nm in pure water (2% DG doped microcavity). At 10nM, 50nM, 100nM, and 500nM, the CWLs of DG are 537.725nm, 536.407nm, 534.818nm, and 531.2899nm, respectively. The wavelength shifts between two adjacent concentrations are shown in table 3.

Figure 2.

Unprocessed data of DG and R6G lasing spectra. The red dashed line indicates the CWL of DG at zero concentration (ddH₂O). The black dashed line indicates the continuous shift of CWL as concentration varies.

Table 3. Wavelength shifts of DG maximum lasing lines (CWL) between two adjacent concentrations

C (nM)	0	10 nM	50 nM	100 nM	500 nM
λ (nm)	539.403	537.725nm	536.407nm	534.818	531.299
$\Delta\lambda$		1.687	1.318	1.589	3.51

Comparing the $\Delta\lambda$ values in table 3 with FSR value shown in figure 1, we can find $\Delta\lambda$ values are much smaller than mode spacing (FSR), indicating that the maximum mode is shifted rather than hoped from one mode to another. The wavelength shift between 100nM and 500nM (3.51nm) is bigger than others because the concentration gap is also big.

So, in fact, the continuous shift of wavelength can be seen at both high and low concentrations. The key of FRET-WGM sensing is the energy transfer between donor and acceptor. In the case of low concentration, although the change of external refractive index is very small, as long as the microcavity is embedded in the acceptor solution and pumped, efficient energy transfer would always occur, which then would result in the blue shift of donor relative to zero concentration. Of course, for the acceptor, the laser will be generated only when the gain is equal to or great than the loss. For the donor, the energy loss will decrease the emission cross-section, resulting in a change in the laser threshold. Through the theoretical analysis (response to reviewer's first comment), we know clearly that there is a one-to-one correspondence relationship between threshold and wavelength. Therefore, for both donor and acceptor, the wavelength shift is a continuous function of concentration.

I am particularly concerned about Fig. 3a in comparison to the added unprocessed data in Fig. S7. First, the authors did not include unprocessed spectra between 0 and 1 μM . Seeing the highly multi-mode spectra spanning over 20 nm, it is not clear at all how the authors were able to separate the lasing spectra of the two dyes (if they are separate) and determine the CWL of each for the low concentrations less than 1 μM at which $\Delta\lambda$ is claimed to be less than 20 nm.

Response: Thanks for the constructive comments. We apologize for not showing the original spectra of 0-1 μm concentration. I understand the reviewer's concern because of the multiple lasing lines of WGM. Here we would like to use the same figure (figure 2 above) to answer this question. Figure 2 shows the unprocessed data of concentration 0-1 μM . We can see that DG and R6G lasing lines at lower concentration are not as many as that of high concentration (actually, even for the high concentrations, as long as the pumping power is controlled just above the threshold, the lasing lines are not as many as we showed in figure S6a of our MS).

For higher concentrations, for example, 500nM and 100nM, at which the DG and R6G spectral envelopes are widely separated, it is very easy to identify the DG lines and R6G lines. For relatively low concentrations, for example, 10nM and 50 nM, we used FSR to help assigning the lasing lines. The measured FSRs for DG and R6G at different concentrations are shown in table 4. We can see that, under certain concentration, the FSRs of DG and R6G are quite distinguishable. The FSR of R6G is always larger than that of DG, which provides a basis for us to distinguish the two spectral envelopes at very low concentration. The decreased FSR of DG can be contributed to the increased refractive index near the sphere surface. The Dragon Green sees a little bit refractive index increase due to the layer of R6G. For R6G, its WGM is pulled outward slightly more than Dragon Green (since the gain position is a little different. The gain for R6G is a little bit more outside than Dragon Green). We have described this in the revised MS (page 9, second paragraph)

Table 4. Measured FSRs of DG and R6G at various concentrations.

	10nM	50nM	100nM	500nM	10 μ M	25 μ M	50 μ M	75 μ M	100 μ M
DG-FSR (nm)	3.803	3.715	3.713	3.429	3.675	3.771	3.443	3.682	3.636
R6G-FSR (nm)	4.367	3.981	3.931	3.881	4.144	4.241	4.141	4.417	4.511

As shown in figure 2, at concentrations of 10nM and 50nM, the spectral envelopes of DG and R6G seem a bit overlapped. Through spectral analysis and FSR calculation, it is found that lasing lines of DG and R6G are obviously different. At 50nM concentration, the FSRs of DG is 3.715nm, while the FSR of R6G is 3.981nm. By measuring the FSRs of two adjacent modes on the spectrum, peaks 1, 2, 3, 4 are assigned to DG lines; while peaks 5, 6 and 7 are assigned to R6G lines. Similarly, we assigned the peaks at concentration of 10nM.

We have included the unprocessed data of concentration 0-500nM in the revised MS. Please refer to figure S6b.

Without adequate feasibility data, I think Fig. 6 should be removed or moved to the supplemental information.

Response: Thanks for the advice. We could move it to the supplementary information.

Finally, I am sincerely grateful to the reviewer for providing insightful comments, which have been of great value in helping us improve the quality of our MS. The significance of our work is mainly two folds: 1) we developed a new strategy to enhance WGM sensitivity. 2) The proposed configuration enables molecular analysis at single cell resolution. Given the difficulties of molecular analysis at single cell resolution (due to the extremely low molecular quantity within a single cell) and importance of single cell analysis (which can reveal cellular heterogeneities and subpopulations), the significance of this work is unquestionable. Our findings will be of particular interest to the life scientist to explore molecular behavior at intracellular level. From this point of view, the data we presented in the MS are original, sufficient and complete.

Replies to Reviewer 2

Reviewer #2 (Remarks to the Author):

The authors reported FRET-based whispering gallery mode (WGM) lasing as a sensing platform, particularly for intracellular sensing.

The microsphere is first doped with donor dyes (such as Dragon Green) and then it is immersed in the acceptor solution such as R6G solution. FRET occurs between the donor dye and the acceptor dye near the microsphere surface. Upon excitation of the donor dye, donor dye lases. When there are acceptor dye molecules near the sphere surface, the FRET is able to pump the acceptor dye for it to lase at a longer wavelength. In the meantime, the donor dye lasing threshold increases (due to the loss to the acceptor dye). With the increased concentration of the acceptor dye, the donor lasing wavelength shifts to a lower wavelength (due to more loss to the acceptor) and meanwhile the acceptor dye lasing wavelengths shifts to a higher wavelength (due to more gain in the acceptor laser system since the acceptor concentration is higher). The spectral gap between the donor lasing wavelength and the acceptor lasing wavelength is then used as the sensing signal. For example, at 1 μM of R6G, the spectral gap between the Dragon Green (donor) lasing wavelength and the R6G (acceptor) lasing wavelength can be 12-20 nm. The authors use the lasing linewidth as the spectral resolution and then estimate the detection limit is about 60 pM of R6G. Finally, the authors use this FRET-WGM sensor to detect the R6G concentration inside cells and 3 μM R6G concentration is experimentally detected.

While the FRET-WGM lasers have been demonstrated many times in the past ten years (see the Refs. 2-5 given below and the authors should cite some of those papers), it is nice to see the authors to use the spectral gap as the sensing mechanism. The authors did thorough experiments to prove the FRET is the mechanism behind the acceptor lasing and is responsible for the spectral gap.

Reponses: Thanks for the valuable comments. We are sorry for not citing Ref.2-5 in our paper. Our proposed approach has fundamental differences with the work in Refs 2-5. We have detailed comparison in response to reviewer's comment 4 below. Nevertheless, we recognize the references mentioned by the reviewer should not be missed in the MS, so we would like to cite the papers in our revised MS (page 14 line 13).

However, there are a few major flaws and drawbacks in this work.

1. The authors introduced a dual core-shell structured ring resonators, one is formed by the microsphere itself and the other one is formed by a layer for R6G molecules near the surface. The evidence is that (1) the free-spectral-range analysis shows the ring diameter for R6G is slightly larger than the ring diameter for Dragon Green. (2) The image. However, the small refractive index contrast between the layer of R6G and the surrounding solution (water + a certain amount of R6G in free solution) may not be able to support another ring resonator. What really matters in the free spectral range calculation is $n \times D$ (refractive index times

diameter). The decreased FSR can be contributed to the increased refractive index near the sphere surface. The Dragon Green sees a little bit refractive index increase due to the layer of R6G. Most of the Dragon Green is still inside the microsphere. For R6G, its WGM is pulled outward slightly more than Dragon Green (since the gain position is a little different. The gain for R6G is a little bit more outside than Dragon Green).

In any case, whether there is a dual-resonator does not affect the sensing mechanism and the sensor performance.

Response: Thanks to the reviewer for the comments and thoughtful instructions. The original dual-cavity was constituted based on the observation of two folds of evidence 1) FSR analysis 2) imaging. It was intended to explain why R6G can lase at the investigated concentration with FRET-WGM setup while in the WGM-only setup R6G can not. I agree with the reviewer that the dual cavity does not affect the sensing mechanism and the sensor performance. The sensing mechanism and the sensor performance were explained in the latter paragraphs of the MS.

Although we have provided evidences, since there are controversial views regarding the dual cavity concept, and I think the reviewer's concern is fair and sensible, we would like to take out the related description and rephrase the text accordingly. Please refer to page 9, the second paragraph. The concept of dual-cavity was derived based on the experimental observation. Taking out the dual-cavity description does not affect the whole structure and any conclusion of the MS, since we have provided adequate evidence to explain the mechanism of R6G lasing.

2. On page 10 (Eq. 1 and the paragraph below). Eq. 1 explains so-called reactive sensing mechanism (or passive sensing). It is applicable to non-FRET-WGM, but cannot be applied to the active sensing. The explanation in the paragraph below Eq. 1 is incorrect (later, the authors use the coupled rate equations to explain the spectral gap between the donor lasing wavelength and the acceptor lasing wavelength – that is correct). Besides, the authors do not give detailed description of $E_x(r_0)$. The plasmonic effect the authors cited in the same paragraph is reactive sensing, to which Eq. 1 is applicable.

Response: we appreciate the reviewer's comments. I would like to say that Eq.1 was not intended to explain our sensing mechanism, it was just a piece of background information. Indeed, Eq.1 is used to explain reactive sensing mechanism. The term "reactive" here refers to the change in spectral properties introduced by a purely reactive interaction, that is, by an elastic scatterer that does not induce any absorption losses. It is in contrast to the term "dissipative sensing mechanism"[ref1]. Any spherical particle interacts with the microcavity, such as molecules or nanoparticles, can be considered as "reactive", while nanorod or nanotube is not. The straightforward solution to boost the sensitivity is to enhance the field $E(r_0)$ seen by the biomolecule [ref2]. For example, by coupling the WGM to plasmonic nanostructures. Here, we demonstrate that coupling FRET to WGM is another way to enhance sensitivity, since energy transfer by FRET can also enhance the field intensity. Apart from the evidence provided in the MS, in our unpublished data, we found that, if energy transfer is reversed, i.e., from outside to inner cavity, the sensitivity is drastically dropped, which supports

our hypothesis in another way.

So, Eq.1 was quoted in the MS just to use as a basis to hypothesize that, at the "binding" site (r_0), since the energy of R6G molecules are increased due to FRET, the sensing capability would be increased. Of course, the mechanism of FRET-WGM is totally different from pure WGM. The observations are different too, for example, a simple redshift of resonance wavelength is a remark for pure WGM; in contrast, wavelength gap between donor and acceptor marks the sensitivity for FRET-WGM. Therefore, we can not apply eq. 1 to explain our results. Instead, we used rate equations to explain the mechanism and absorption/emission cross sections to explain concentration-dependent wavelength gap (as advised by the reviewer).

Thanks to the reviewer's comments, we have rephrased the text to make the description more clear and logic, please refer to page 9-10 in the revised MS.

Ref1. Y Zhi, XC Yu, Q Gong, Lan Yang, and Yun-Feng Xiao, Single Nanoparticle Detection Using Optical Microcavities. *Adv. Mater.* **2017**, 29, 1604920

Ref2. M. A. Santiago-Cordoba, S. V. Boriskina, F. Vollmer, M. C. Demirel, *Appl. Phys. Lett.* **2011**, 99, 073701.

3. The mechanism behind the spectral gap between the donor lasing and acceptor lasing is well known. An earlier paper was published in 2000 (see Ref. 1 below). Later, there are a few more papers published to explain in details the FRET laser and the lasing wavelength in the FRET laser (see Refs. 2-5). While the rate equation is nice, but simple absorption and emission cross section analysis may be more intuitive. In short, when the acceptor concentration increases, the donor loss increases. The absorption/emission cross section analysis shows the shorter wavelength is easier to lase. That's why the donor lasing wavelength shifts towards the shorter wavelength. Similarly, when the acceptor concentration increases, the acceptor gain becomes higher. The same absorption/emission cross section analysis shows the acceptor lasing becomes easier at a longer wavelength. That's why the acceptor lasing wavelength shifts to the longer wavelength. Therefore, a few more citations should be included to cover the FRET lasing and the lasing wavelength shift mechanism (for example, Refs. 1-5).

Response: we appreciate the reviewer's comments. We would say that, since the configuration of FRET-WGM in our MS is different from the earlier papers (ref2-5), our mechanism is also different from previous work.

To address the reviewer's concern, we compared our work with previous work (ref2-5). We would say that their core concerns and concept are fundamentally different from ours.

(1) The format of energy transfer is totally different and thus the mechanisms are different. In their cases, donor and acceptor are assembled via a DNA structure and then placed in a micro-ring cavity. So, energy transfer is within the DNA molecule. The total energy of the microcavity remains unchanged. The effect of FRET on sensitivity was not specially studied in Ref2-5 since their ultimate goal was not to increase sensing ability. In our case, donor is doped

in the microcavity that is embedded in acceptor surrounding. This way, the energy transfer occurs from inner-cavity to extra-cavity, resulting energy loss within the cavity and energy gain of the outer cavity molecules. Therefore, FRET was able to increase the sensing ability by 4 orders of magnitude compared to the non-FRET counterpart.

(2) The objectives are completely different. They use FRET to study DNA structure or to extend the lasing wavelength, they were not aiming to sensing application. We use FRET to enhance WGM sensitivity to enable intracellular applications.

(3) The application scenarios are different. The shape (micro-ring) & dimension of their cavity as well as the optofluidic configuration determines that their setup is not be applicable for intracellular work. We use a single microsphere cavity that can be internalized by a cell enabling intracellular analysis.

Our findings demonstrate that FRET and WGM in the proposed configuration complement each other in a mutual way. WGM enhances energy transfer efficiency of FRET; in turn, FRET enhances WGM sensitivity. They play a part together to achieve high sensitivity and enable accomplishment of intracellular sensing.

[Image of Figure 1 Redacted]

Figure 1. Comparison of our work with previous work of refs 2-5 mentioned by the reviewer.

We appreciate the reviewer's advice to use absorption and emission cross section analysis to explain the wavelength shift. To explain the wavelength shift of donor and acceptor, we used

quasi-3 level system, which is also related to absorption and emission cross sections of donor and acceptor. Considering the focus of this work is to demonstrate the feasibility of the proposed approach for intracellular applications, we did not go into detailed analysis of the absorption and emission cross sections. Following the reviewer's advice, we have added a brief description of the concentration-dependent wavelength shift using analysis absorption and emission cross section analysis, please refer to page 15, the last sentence of 2nd paragraph in the revised MS, and section 4 of SI. Here I would like to briefly outline how we would use rate equation and cross section analysis to quantify the wavelength shift:

As we described in the revised MS, the acceptor-based laser rate equation is shown in Eqn. (1):

$$\frac{dq_a(t)}{dt} = \frac{cq_a(t)n_a}{n_2} \sigma_{ea}(\lambda) - \frac{cq_a(t)(N_a - n_a)}{n_2} \sigma_{aa}(\lambda) - \frac{q_a(t)}{\tau_{ea}} \quad (1)$$

Under steady-state conditions, Eqn. (1) can be written to Eqn. (2):

$$n_a \sigma_{ea}(\lambda) - (N_a - n_a) \sigma_{aa}(\lambda) - \frac{2\pi n_2}{\eta Q \lambda} = 0 \quad (2)$$

Therefore, from Eqn. (2), we can obtain the fraction of acceptor molecules at the excited state under the threshold condition [ref1]:

$$\gamma_{tha} = \frac{n_a}{N_a} = \frac{1}{\sigma_{ea}(\lambda) + \sigma_{aa}(\lambda)} \left[\sigma_{aa}(\lambda) + \frac{2\pi n_2}{\eta Q \lambda N_a} \right] \quad (3)$$

Where γ_{tha} represents the lasing threshold of acceptor dye.

In addition, we have derived donor-based equations to describe the wavelength-concentration relationship. Please not that, as far as we know, the relationship between donor wavelength shift and concentration has never been discussed in previous publications. The donor-based laser rate equation can be described in Eqn. (4).

$$\frac{dq_d(t)}{dt} = \frac{cq_d(t)n_d}{n_1} \sigma_{ed}(\lambda) - \frac{cq_d(t)(N_d - n_d)}{n_1} \sigma_{ad}(\lambda) - \frac{q_d(t)}{\tau_{ed}} + \frac{cq_d(t)n_a}{n_2} \sigma_{ea}(\lambda) - \frac{cq_d(t)(N_a - n_a)}{n_1} \sigma_{aa}(\lambda) \quad (4)$$

Where N_d is the total concentration of donor molecules, and n_d is molecular density of donor dyes in the excited state. $\sigma_{ed}(\lambda)$ and $\sigma_{ad}(\lambda)$, respectively, are the donor emission and absorption cross-section. q_d is the donor emitted photon density. τ_{ed} is the lifetime of the photons in the cavity, which equals to $\eta Q \lambda / 2\pi c$. n_1 and c represent the refractive index of the microcavity (~ 1.59) and light speed in the vacuum, respectively. η means the fraction of mode energy in the evanescent field. Under steady-state conditions, Eqn. (4) can be written to Eqn. (5):

$$n_d \sigma_{ed}(\lambda) - (N_d - n_d) \sigma_{ad}(\lambda) - \frac{2\pi n_1}{\eta Q \lambda} + \frac{n_1 n_a}{n_2} \sigma_{ea}(\lambda) - (N_a - n_a) \sigma_{aa}(\lambda) = 0 \quad (5)$$

From Eqn. (5), we can obtain the fraction of donor molecules at the excited state under the threshold condition:

$$\gamma_{thd} = \frac{n_d}{N_d} = \frac{1}{\sigma_{ed}(\lambda) + \sigma_{ad}(\lambda)} \left[\sigma_{ad}(\lambda) + \frac{2\pi n_1}{\eta Q \lambda N_d} + \frac{(1 - \gamma_{tha}) \sigma_{aa}(\lambda) N_a}{N_d} - \frac{\gamma_{tha} N_a \sigma_{ea}(\lambda)}{n_2 / n_1} \right] \quad (6)$$

Where γ_{thd} represents the lasing threshold of the donor dye.

According to Eqn. (3) and (6), different R6G acceptor concentrations will give to corresponding γ_{th} values of donor and acceptor at respective wavelengths. So when the pump energy reaches the threshold (γ_{th}), donor or acceptor will lase at a particular wavelength. Since the absorption and emission cross sections are determined by the dye concentration as well as the gain or loss via FRET energy transfer, also, the emission and absorption cross sections are a continuous function of wavelength, thus, the thresholds will be a function of the wavelength (other parameters can be regarded as fixed values). Therefore, under different acceptor concentrations in which energy transfer is different, the laser threshold and resonance wavelength will be different, i.e. the threshold and resonance wavelength is dependent on the acceptor concentration. Therefore, theoretically, we could simulate and calculate the concentration-dependent shift of the resonance wavelengths of both donor and acceptor. The relationship between concentration and wavelength gap would then be constructed. The theoretical model will be of great significance in predicting the donor-acceptor wavelength gap at a certain acceptor concentration and thus theoretically explain the sensing mechanism.

The reasons that we haven't included the aforementioned analysis to the MS include: **(1)** using absorption and emission cross section to analyze the wavelength shift can not be accomplished at one stroke. It involves measurements of the absorption and emission cross sections at various concentrations, determining energy transfer efficiency under different combinations of donors and acceptors and lots of calculations and analysis. **(2)** the major focus of this MS is the demonstration of intracellular application. As far as we know, this will be the first case of WGM sensing for intracellular analysis at molecular level. Previously, there have been several exciting reports (*Nat Photonics*. 2015 9(9):572; *Nat Photonics*. 2019 13(10):720; *Light Sci Appl*. 2021 25;10(1):23; *Nat Commun*. 2018 16;9(1):4817; *Nat Photonics*. 2020 14 (7) ,452) showing intracellular WGM applications. Their work is mainly focused on cellular behavior, such as tracking cell movement via WGM labeling, monitoring subtle changes of intracellular microenvironment etc., i.e., more focused on information acquisition at cellular level. We are very happy to complete the job and publish the data somewhere else. For this MS, we think the current data are original, sufficient and complete.

Ref1. Chen, Y.-C.; Chen, Q.; Fan, X., Optofluidic chlorophyll lasers. *Lab Chip* 2016, 16 (12), 2228-2235.

4. The authors claim that 60 pM R6G detection limit is possible. This is misleading. First, this FRET-WGM sensing system is not linear and the slope is very shallow. For example, at 1 uM of R6G (Table 1), the spectral gap is 12.8 nm. At 0.01 uM, the spectral gap is 6.15 nm. The concentration drops by 100 fold, but the spectral gap decreases only 50%.

Response: we really appreciate the reviewer's constructive comments. Follow the reviewer's advice to plot Figure 3b and c in the log-log scale (please see reviewer's point 4), we have replotted the curves and also modified the MS accordingly. Indeed, the relationship between wavelength gap and concentration is not linear.

Using replotted curves and simulated equations, we re-calculated the limit of detection (still use the line width as limit of detection). The LOD for FRET-WGM is 15.2nM; and for pure

WGM is 747.4uM. The difference is still four orders of magnitude, which is around 5×10^4 . We have modified our MS, please refer to page 11 line 2, page 12 line 12.

Many thanks to the reviewer indeed.

Second, there is a concentration floor for R6G below which no lasing can be achieved. Without lasing, there is no spectral gap. The lower floor of R6G is on the order of 1-10 uM.

Response: Thanks for the comments. I understand the reviewer's concern and would like to explain this in detail. If there is a concentration floor, there will be two cases determining the floor: 1) a concentration at which R6G cannot lase or 2) a concentration at which although R6G lases, the CWLs of DG and R6G can not be resolved. For the first case, because of the WGM cavity, energy transfer by FRET is much more efficient than without WGM, thus, R6G molecules can gain substantial energy. We observed that, even at very low R6G concentration (10 nM), R6G was able to lase easily. In theory, as long as R6G molecules gain enough energy, R6G can always lase, although this may not be achievable experimentally due to limitations of experimental condition. So, for the first case, the concentration floor is not so certain, or predicable. For the latter case, as long as there is energy transfer, there is always blue-shift of DG lasing line. Whether $\Delta\lambda$ can be measurable depends on the line width and the resolution of the spectrometer. This is why we defined the limit of detection using line width. Indeed, at very low concentration, the DG and R6G spectral envelopes seem a bit overlapped, as shown in figure2 below. However, with the help of FSR analysis, we can assign the DG and R6G lasing lines without ambiguity.

Figure 2. Unprocessed data of DG and R6G lasing spectra. The red dashed line indicates the CWL of DG at zero concentration (ddH₂O). The black dashed line indicates the continuous shift of CWL as concentration varies.

Figure 2 shows the unprocessed data of concentration 0-1 μ M. We can see that DG and R6G lasing lines at lower concentration are not as many as that of high concentration (actually, even for the high concentrations, as long as the pumping power is controlled just above the threshold, the lasing lines are not as many as we showed in figure S6 of our MS).

For higher concentrations, such as 500nM and 100nM, at which the DG and R6G spectral envelopes are widely separated, it is very easy to identify the DG lines and R6G lines. For relatively low concentrations, for example, 10nM and 50 nM, we used FSR to help assigning the lasing lines. Under certain concentration, the FSRs of DG and R6G are quite distinguishable. The FSR of R6G is always larger than that of DG, which provides a basis for us to distinguish the two spectral envelopes at very low concentration.

As shown in figure 2, at concentrations of 10nM and 50nM, the spectral envelopes of DG and R6G do not seem to be separated at a glance. Through spectral analysis and calculation of FSR, it is found that lasing lines of DG and R6G are obviously different. At 50nM concentration, the FSRs of DG is 3.715nm, while the FSR of R6G is 3.981nm. By measuring the FSRs of two adjacent modes on the spectrum, peaks 1, 2, 3, 4 are assigned to DG lines; while peaks 5, 6 and 7 are assigned to R6G lines. Similarly, we assigned the peaks at concentration of 10nM.

We have included the data in revised MS, please refer to SI, figure S6.

Third, there is a floor for the spectral gap. For example, the lowest lasing wavelength may be, say, 532 nm. You cannot push the spectral gap below a certain number, for example 1 nm. I guess

Response: Thanks for the comments. We agree with the reviewer that there is a concentration floor. This floor would be determined by the resolution of the spectrometer. Let's still use figure 2 to explain. At zero concentration (pure ddH₂O), the DG lasing is @539.403nm (2% DG doped microcavity). In the presence of R6G, the overall effects of FRET & WGM render DG lasing line blueshift compared to zero concentration (as long as there is energy transfer, there is always a blueshift of DG lasing line); meanwhile render R6G lasing. As we analyzed above using rate equations and absorption and emission cross sections, the CWL wavelength gap between DG and R6G is a continuous function of acceptor concentration. At extremely low concentration, although there is still blueshift compared to 0 nM, the CWL of DG would be very close to the CWL of zero concentration. At this point, the CWL of R6G will also be very close to the DG CWL. Therefore, as long as R6G can lase, the floor is determined by the resolution of the spectrometer. In our case, the resolution of the spectrometer (0.04nm), and line width is 0.25nm. For the sake of reliability, we use line width as the limit of detection.

For clarity, it is suggested the authors to plot Figure 3b and c in the log-log scale.

Response: thanks for the constructive comments, we have re-plotted the figures and rephrased the text accordingly. Indeed, log-log scale curves make more sense. We have by replaced the figures in modified MS, please refer to figure 3b, c and figure 4d.

Figure 3. Concentration dependent wavelength (left panel) and wavelength gap curves plotted using log-log scale. Left and right panel are replotted figure 3b and c, respectively.

5. Relying on the electrostatic interaction to pull more R6G towards the sphere surface is a little risky, as it affects the quantitation of the R6G concentration (or any other analyte concentration). In reality, if the ionic strength in solution changes, the R6G surface density may change quite a bit.

Response: we appreciate the reviewer's insightful comments. Electrostatic interaction might be a major force for R6G adsorbed on the microcavity surface. We can not rule out other interactions such as hydrogen bonds etc. Anyhow, it is adsorption force between R6G and microcavity. In the world, there are plenty of applications using adsorption for quantitative analysis, for examples, chromatographic method for molecular separation; DNA molecules adsorbed on the surface of graphene or other 2D materials for sensing (*Nat Commun.* 2019 10:28 <https://doi.org/10.1038/s41467-018-07947-8>); also antibody and antigen interactions (electrostatic, hydrogen bond and hydrophobic interaction), and so forth. All the mentioned adsorption interactions are commonly used for quantitative analysis. Under certain condition, the adsorption energy is a fixed value, thus the quantification is reliable. We agree with the reviewer that, if the conditions change, the interaction may change quite a bit. So, there is always a definite condition for a designated experiment. As long as the condition remains same, the results are reproducible. For our MS, there have been 4 independent individuals working on the same experiment with a time-span of five year, they all got same results, indicating reliability of the rationale.

Moreover, this work is just a demonstration of the feasibility to intracellular application. In real world, this configuration would not be directly used for intracellular analysis. For example, if we want to detect intracellular microRNA molecules, we could use R6G or other fluorescent dye molecules to label the probe (DNA) via chemical reaction and hybrid with a recognizing element (DNA) chemically linked on the microcavity surface. In this way, R6G molecules are seated on the microcavity surface quantitatively forming a FRET-WGM shape. In the presence of analyte, the R6G labeled probe will be replaced by the analyte and result in a wavelength shift of the lasing spectrum, which can be used for identification and quantification.

Finally, I am sincerely grateful to the reviewer for providing insightful comments, which have been of great value in helping us improve the quality of our MS. The significance of our work

is mainly two folds: 1) we developed a new strategy to enhance WGM sensitivity. 2) The proposed configuration enables molecular analysis at single cell resolution. Given the difficulties of molecular analysis at single cell resolution (due to the extremely low molecular quantity within a single cell) and importance of single cell analysis (which can reveal cellular heterogeneities and subpopulations), the significance of this work is unquestionable. Our findings will be of particular interest to the life scientist to explore molecular behavior at single cell level. From this point of view, the data we presented in the MS are original, sufficient and complete.

1. Moon et al., Phys. Rev. Lett. Vol. 85, 3161 (2000).
2. Sun et al., PNAS Vol. 107, 16039 (2010).
3. Sun et al., Angew. Chem. Intl. Ed. Vol. 51, 1236 (2012)
4. Chen et al., Lab Chip Vol. 13, 3351 (2013)
5. Chen et al., Lab Chip. Vol 16, 2228 (2016)

Reviewer 3: No further questions

Reviewers' comments:

Reviewer #1 (Remarks to the Author):

The authors maintain their opinion that the CWL is a "continuous" function of concentration and provided a dataset as the evidence. I remain doubtful about this claim. I had hard time understanding the authors' argument. It appears to argue that the laser mode wavelength is determined by concentration-dependent lasing threshold, but this is not correct or at least too oversimplified. I argue that the laser mode wavelengths should always coincide with (part of) the discrete resonance modes of the cavity. The gain/loss profile simply determines the relative intensity (including whether they are above or below lasing threshold) of the cavity modes.

Looking at the data provided as Fig. 2 in the Response letter, I assume that the particular cavity support both TE and TM modes as I see small peaks between larger peaks separated by the FSR of ~ 3.9 nm. The data show that it is the central (TE) mode at ~ 539 nm that has the maximum intensity in water. At 10 nM of R6G, the central mode becomes considerably weaker, and instead the neighboring (TM) mode separated by ~ 1.7 nm lases. At 50 nM, this mode becomes small, and another (TE) mode separated from the original (TE) mode by one free-spectral range takes the maximum intensity. This mode "hop" continues as the concentration increases. I don't have good explanation why the maximum-intensity mode should alternate between TE to TM modes, but this is my best interpretation of the data in Fig. 2.

The authors claim that the CWL is a continuous shift, rather than the jump from one mode to another. I'd like to see experimental data that support this. What happens between 10 and 50 nM? Do the modes really shift continuously? If so, this on its own is the most surprising, significant result of this paper. And then what's a clear, physically intuitive explanation for this potentially novel phenomenon?

If my interpretation is correct, then there would be several points in this manuscript that need to be corrected. The most important point is about the limit of detection (LOD). According to my argument, LOD will come from the FSR (3.9 nm) rather than the much smaller laser line-width. (In the data in Fig. 2, the wavelength change between concentrations are about a half of the FSR, but this could be simply because the size of the cavity is such that the TE and TM modes are positioned in the middle of each other. In general, their positions can be arbitrary depending on the size of the cavity.) Along with this point, I see the large error bars in Fig. 3(b), which appears to be in the order of FSR (~ 3.9 nm) indeed.

Minor, it will be helpful to add more y-axis ticks in the log-log plots of Δ_{λ} (e.g. at 5, 10, 20, 40 nm...).

Reviewers' comments:

Reviewer #1 (Remarks to the Author):

1. The authors maintain their opinion that the CWL is a "continuous" function of concentration and provided a dataset as the evidence. I remain doubtful about this claim. I had hard time understanding the authors' argument. It appears to argue that the laser mode wavelength is determined by concentration-dependent lasing threshold, but this is not correct or at least too oversimplified. I argue that the laser mode wavelengths should always coincide with (part of) the discrete resonance modes of the cavity. The gain/loss profile simply determines the relative intensity (including whether they are above or below lasing threshold) of the cavity modes.

Thanks to the reviewer for the comments.

First, I would like to clarify that there are two different types of mode "hop" we are discussing:

- (1) Mode competition within the same mode format, i.e. either TM or TE, between different mode numbers, e.g., between TE_{133} and TE_{134} . In this case, if there is mode "hop", the hop value should be FSR (3.9nm) or FSR multiplied by an Integer. As we discussed in the last response letter, only when the pumping energy is higher enough, i.e. higher than the saturation gain, there are mode competition phenomena (see our Fig S1a). For example, the original dominant mode TE_{134} may become less dominant, while the original less dominant TE_{133} may become dominant. In our experiments, under the investigated conditions, i.e. pumping energy is just above the threshold (since the final application is in cell, we tend to keep the pumping energy as low as possible), therefore, there is only one dominant mode number (i.e. the CWL). We have discussed this issue in the last response letter. .
- (2) Mode competition between TE and TM of same mode number, for example, TE_{133} and TM_{133} . I guess this is the major concern of the reviewer in this time's comment letter.

Generally speaking, in active WGM, TE modes have typically higher Q factors than TM modes (we cannot say this applies to all cases) and for this reason, their lasing threshold is lower, where TM modes are discernible as small peaks only, and thus presumably not lasing under the conditions. It is widely accepted that WGM with the same mode number red shift as the refractive index increases. This is the foundation for WGM sensing. It is unlikely that, under same pumping conditions, at one concentration the dominant mode is TE and another concentration is TM. [please see Ref1-3, Ref 1. APPLIED PHYSICS LETTERS **94**, 031101 2009; Ref 2 Annals of Biomedical Engineering, Vol. 37, No. 10, October 2009.; Ref 3. Nature photonics, 2020 <https://doi.org/10.1038/s41566-020-0631-z>]

In our case, the sensing mechanism is different from pure active WGM. Refractive index change is not a dominant factor, instead, FRET regulated energy transfer is the dominant factor. Even so, it is still true that the lasing threshold of TE mode is much lower than TM mode, thus it favors for TE mode lasing. In the FRET-WGM case, the lasing threshold is a function of the transferred energy. Since the transferred energy via FRET is a continuous

function of the concentration, it is not difficult to understand that the mode shift is also a continuous function of concentration. This is proved by our experimental data, as discussed in the text below (response to reviewer's point 2,3 and 4).

To prove the above statement, we constructed a theoretical model via analysis of the absorption/cross section and fluorescence life time. According to the equations we provided in the last response letter, different acceptor concentrations will give rise to different γ_{th} values at corresponding wavelengths (SI Fig. S15). When the pump energy reaches specific molecules fraction (γ_{th}), lasing will occur at the particular wavelength (the lowest point in the curve of SI Fig. S15e & f). Therefore, there is a relationship between the mode wavelength and the concentration-dependent lasing threshold.

When our manuscript was under review, a paper was published in *Nanoscale* (Z. Yuan, X. Tan, X. Gong, C. Gong, X. Cheng, S. Feng, X. Fan and Y. Chen, *Nanoscale*, 2021, DOI: 10.1039/D0NR07921A.). In this paper, WGM and FRET are integrated for biosensing application. The authors of the paper demonstrated a continuous red-shift of the maximum lasing mode of the acceptor molecule in response to increasing ratio of A/D (acceptor/donor), and explained the mode shift mechanism via simulating dose dependent wavelength shift of the acceptor molecule. In addition, they also use the dominant mode shift, i.e. the maximum lasing peak, as a measure for sensing, which is similar to our description. They also use similar equations to predict the wavelength shift in response to concentration change. This paper would more or less support our statement in regarding to the concept of CWL, mode shift and relationship between mode shift and concentration. Of course, the major focus of our work is different from their work. The major differences are: (1) we realized donor and acceptor dual lasing and thus enhanced sensing limit. (2) we not only constructed the wavelength-concentration relationship for the acceptor molecule, also constructed the wavelength-concentration relationship for the donor molecule, which has not be reported previously by any paper. (3) for the first time, we applied FRET-WGM for intracellular analysis at molecular level and single cell resolution.

2. Looking at the data provided as Fig. 2 in the Response letter, I assume that the particular cavity support both TE and TM modes as I see small peaks between larger peaks separated by the FSR of ~ 3.9 nm. The data show that it is the central (TE) mode at ~ 539 nm that has the maximum intensity in water. At 10 nM of R6G, the central mode becomes considerably weaker, and instead the neighboring (TM) mode separated by ~ 1.7 nm lases. At 50 nM, this mode becomes small, and another (TE) mode separated from the original (TE) mode by one free-spectral range takes the maximum intensity. This mode "hop" continues as the concentration increases. I don't have good explanation why the maximum-intensity mode should alternate between TE to TM modes, but this is my best interpretation of the data in Fig. 2.

Thanks for the comments.

Based on our data, from one concentration to another concentration, the distance between TE and TM ($\lambda_{TE} - \lambda_{TM}$) almost remains same as showed in the following table 1, however, the shift of the resonant wavelength is variable (can be any value depend on the concentration)

and the shifted value ($\Delta\lambda$) is not equal to the “hop” value ($\lambda_{TE} - \lambda_{TM}$), which suggests that it is shift rather than hop. Let's take the donor mode shift as an example as shown in figure 1 below (same as figure 2 in the last response letter), $\Delta\lambda$ of CWLs between 50-100nM is 1.8586, however, $\lambda_{TE}-\lambda_{TM}$ is 1.4758nm, which is different from the $\Delta\lambda$ value. The $(\lambda_{TE}-\lambda_{TM})$ value is basically always around 1.5nm. We can measure any $\Delta\lambda$ value between two concentrations, they are variable depend on the concentration (we know λ -c relationship is not linear) and usually not equal to $(\lambda_{TE}-\lambda_{TM})$. We also performed more experiment via including a concentration (25nM) in between two adjacent concentrations (10 and 50nM), the data are shown in the text below (response to reviewer's point 3). In addition, we also analyzed our previous data of higher concentration (500nM-10uM), which is also included in the text below (please see response to reviewer's point 3).

Figure 1. Lasing emission of DG and R6G at 0-500nM R6G concentrations.

3. The authors claim that the CWL is a continuous shift, rather than the jump from one mode to another. I'd like to see experimental data that support this. What happens between 10 and 50 nM? Do the modes really shift continuously? If so, this on its own is the most surprising, significant result of this paper. And then what's a clear, physically intuitive explanation for this potentially novel phenomenon?

Thanks for the comments. Following the reviewer's advice, we performed additional experiments and added data of concentration 25nM, as shown in figure 2 below. We also summarized the data of 0-100nM in table 1 below. We can see that from 10nM to 25nM, the

$\Delta\lambda$ value is 0.699nm, from 25nM to 50nM, the $\Delta\lambda$ value is 0.619nm. The $\Delta\lambda$ values of two adjacent concentrations are not same as the $(\lambda_{TE}-\lambda_{TM})$ value, suggesting mode shift instead of mode hop. We have listed the values of $\Delta\lambda$ of two adjacent concentrations and $(\lambda_{TE}-\lambda_{TM})$ values in table 1. Please note that the resonance wavelength values in the table are average of 10 microsphere cavities, the errors are very small as shown in the table.

Figure 2. Lasing emission of DG and R6G at 10-100nM R6G concentrations.

Table 1. Summary of resonance wavelength, TE-TM, and errors of different R6G concentration.

C	0	10nM	25nM	50nM	100nM
λ	539.403nm	537.725nm	537.026nm	536.407nm	534.818nm
$\Delta\lambda=\lambda_{c2}-\lambda_{c1}$	0	1.678	0.699	0.619	1.8568
TE-TM	1.593259	1.517330	1.563509	1.49888	1.47683
误差	0.180531	0.32938	0.212161	0.27656	0.13578

The above data are in the very low concentration range. In order to have a full view, we also performed additional experiments of higher concentrations using 2% DG doped microsphere, as shown in figure 3 and table 2 below. The results are similar to that of 1% DG microsphere presented in the MS (SI, figure S6a). From 500nM to 1uM, the $\Delta\lambda$ value is 0.685nm; from 1uM to 5uM, the $\Delta\lambda$ value is 1.06nm; from 5uM to 7.5uM (very small concentration change), the $\Delta\lambda$ value is 1.696nm; from 7.5uM to 10uM (again very small concentration change), the $\Delta\lambda$ value is 1.969nm. Again, the $\Delta\lambda$ value is not related to the $(\lambda_{TE}-\lambda_{TM})$ value. If it is a mode "hop"

between TE and TM, the $\Delta\lambda$ values would be either similar to the $(\lambda_{TE}-\lambda_{TM})$ value or multiplied by an integer. In fact, this is not the case in our experiments. If we take another careful look of the data, we can find a steep-slope at this concentration range (1-10 μ M), please see figure 3b green line, i.e. small concentration change corresponds to big wavelength shift. This is in contrast to the low concentration data. Understandably, since the relationship between wavelength and concentration is not linear, $\Delta\lambda$ values are not same in between two concentrations. We only chose a few concentrations to perform the experiments in order to obtain enough data for plotting a full curve. It is unlikely to acquire data of all continuous concentrations. Hope our data can help the review to understand our statement.

Figure 3. Lasing emission of DG and R6G at 500nM-10 μ M R6G concentrations (the experiments were performed using 2% DG doped microsphere).

Table 2. Summary of resonance wavelength, TE-TM, and errors of different R6G concentration

C	500nM	1 μ M	5 μ M	7.5 μ M	10 μ M
λ	531.297	530.612nm	529.552nm	527.856nm	525.887nm
$\Delta\lambda=\lambda_{c2}-\lambda_{c1}$	0	0.685	1.060	1.696	1.969
TE-TM	1.47119	1.520172	1.593214	1.574559	1.537788
Errors	0.28994	0.225719	0.167895	0.14292	0.20431

4. If my interpretation is correct, then there would be several points in this manuscript that need to be corrected. The most important point is about the limit of detection (LOD). According to my argument, LOD will come from the FSR (3.9 nm) rather than the much smaller

laser line-width. (In the data in Fig. 2, the wavelength change between concentrations are about a half of the FSR, but this could be simply because the size of the cavity is such that the TE and TM modes are positioned in the middle of each other. In general, their positions can be arbitrary depending on the size of the cavity.) Along with this point, I see the large error bars in Fig. 3(b), which appears to be in the order of FSR (~3.9 nm) indeed.

Thanks for the comments.

Somehow I understand the reviewer's point in regarding the definition of limit of detection (LOD). The reviewer suggested to use the mode space (FSR) instead of line width ($\delta\lambda$) as the limit of detection. The reason for that could be because of the reviewer's "mode hop" standpoint. If the reviewer can accept the mode shift explanation, it would be easy to understand why we use line width ($\delta\lambda$) instead of FSR as LOD. In WGM sensing, line width ($\delta\lambda$) is widely accepted as a measure of LOD as described by Prof Vollmer. (Vollmer, F. & Arnold, S. Whispering-gallery-mode biosensing: label-free detection down to single molecules. *Nature Methods*, 5(7), 591-596 (2008)) and other authors (e.g. Ref1-3 above). Since, essentially, it is a wavelength shifting in our experiments, using line width ($\delta\lambda$) as a limit to resolve two adjacent peaks is quite understandable. Moreover, in any case, as long as we use a unified standard, the LOD of FRET-WGM is always four orders of magnitude enhanced than the pure active WGM, this is for sure.

In response to reviewer's concern of error bars in fig. 3(b), we have presented the error bar values here (The errors in different lines from top to bottom correspond to concentrations from low to high in Fig. 3(b)):

For DG		For R6G	
Errors	2X errors	Errors	2X errors
0.26952	0.53904	0.26952	0.53904
0.40035	0.8007	0.51368	1.02736
0.97822	1.95644	0.4316	0.8632
0.86640	1.7328	1.0748	2.1496
1.02331	2.03106	0.45338	0.90676
1.13968	2.27936	0.80133	1.60267
0.73917	1.47834	0.81261	1.62521
0.90529	1.81057	0.65699	1.31399
0.55874	1.11748	0.42789	0.85578
0.53044	1.06088	0.67286	1.34572
1.01817	2.03634	0.65994	1.31988
0.29884	0.59768	0.5049	1.0098
0.57202	1.14404	0.6289	1.2578
0.53889	1.07778	0.56673	1.13346
0.25834	0.51668	0.29044	0.58088
0.59806	1.19612	0.78373	1.56746
1.23441	2.46882	0.87283	1.74566

Some of the 2X error values (yellow highlight) may coincidentally similar to the $(\lambda_{TE}-\lambda_{TM})$ value (around 1.5nm). However, most of the 2X error values are quite different from the $(\lambda_{TE}-\lambda_{TM})$ value, which is understandable because the errors are random. (I'm a bit confused when reading the reviewer's comments, the last sentence in the paragraph by saying "in the order of FSR (~3.9 nm) indeed", does the reviewer mean $(\lambda_{TE}-\lambda_{TM})$ rather than FSR (3.9nm)? if the reviewer really means FSR, then the 2X error values are far lower than the FSR(3.9nm) values.)

5. Minor, it will be helpful to add more y-axis ticks in the log-log plots of Delta_lambda (e.g. at 5, 10, 20, 40 nm...).

Thanks for the advice, we have modified the figures by adding more ticks, please refer to Figure 3b, c, figure 4d and figure 5h in the revised MS and figure S9c in SI.

I really appreciate the reviewer's comments, especially the professional attitude in evaluation of our manuscript. We benefited from the deep discussion with the reviewer. The reviewer's comments not only greatly improved the quality of our MS, also provoke us to think from a different angle, which is very critical for a scientific researcher.

Finally, I would like to take the opportunity to address the significance of our work again after being through four rounds of review with time-span of one and half years.

(1) The most significance of our work is demonstration of the feasibility of FRET-WGM for intracellular sensing at single cell resolution. More importantly, it is live cell, real-time, molecular level and quantitative analysis, which will be the first case in the area of WGM microcavity sensing. Given the difficulties of molecular analysis at single cell resolution (due to the extremely low molecular quantity within a single cell) and importance of single cell analysis (which can reveal cellular heterogeneities and subpopulations), the significance of this work is unquestionable.

(2) Secondly, for the first time, we demonstrate that introducing FRET to WGM can dramatically increase the sensing performance (This lays the fundamental for the technique to be used for intracellular applications). Previously, scientists in the WGM sensing area have developed a number of techniques to increase the sensitivity to allow ultrasensitive detection and remarkable success has been achieved (Nature Photonics 2016 10, 733-739; Nano Letters 2013 13(7), 3347-51; Light: science & Applications 2016 5(1), e16001; Nature 2017 548, 192-196). However, none of them reported FRET could enhance the sensing performance of WGM (in the abovementioned *nanoscale* paper, they only achieved acceptor lasing). Furthermore, although detection limit has been pushed down to single particle, the previous setups are not applicable for intracellular study because of the requisite of optical coupler (fiber or prism). Our finding provides a new approach to greatly enhance the WGM sensitivity by 4 orders of magnitude and meanwhile to enable intracellular quantitative analysis.

REVIEWERS' COMMENTS

Reviewer #2 (Remarks to the Author):

None.